

# Salinity Trends and Mass Balances in the Mediterranean Sea: The Role of Air-Sea Freshwater Fluxes and Oceanic Exchange

Chao Liu[1], Xinfeng Liang[2], Lisan Yu[1]

[1] Department of Physical Oceanography, Woods Hole Oceanographic Institution, Woods Hole, Massachusetts, USA
[2] School of Marine Science and Policy, University of Delaware, Lewes, DE, USA

*Correspondence to*: Chao Liu  (chao.liu@whoi.edu)

**Abstract.** Understanding the drivers of salinity and mass variability in the Mediterranean Sea is essential for assessing regional climate impacts and elucidating climate-driven changes in the water cycle. Although it is possible to close the Mediterranean mass and salinity budgets within uncertainty ranges, the relative contributions of key boundary fluxes, namely, surface

freshwater fluxes (Evaporation-minus-Precipitation-minus-runoff) and water exchanges through the Strait of Gibraltar, remain unclear. To address this, we analyzed the Mediterranean's mass and salinity budgets of 2003-2017 using ECCOv4r4. Our findings reveal a delicate balance between boundary fluxes that jointly regulate the Mediterranean's mass and salinity dynamics. Surface freshwater fluxes play an essential but often understated role in modulating salinity through changes in volume: salinity decreases when precipitation adds volume, and increases when evaporation reduces volume. This insight

further clarifies how the inflow of relatively fresh Atlantic water (AW) and the outflow of saltier Mediterranean water (MOW) can, counterintuitively, lead to an overall reduction in salinity. We found that surface fluxes, primarily driven by evaporation, account for most salinity variability, contributing approximately 1.80±0.10 Sv. The Gibraltar exchange is critical for maintaining mass balance, adding 0.30±0.20 Sv of salt. However, due to density differences between AW, MOW, and Mediterranean water, a net salinity reduction of -1.48±0.20 Sv is achieved through the Strait. This results in an overall increase

of 0.29±0.09 Sv in salinity over the 15-year period, consistent with the salinification trend reported in previous studies. These findings provide a more comprehensive perspective of regional water cycle dynamics.

## 1 Introduction

The Mediterranean Sea is known to be sensitive to climate change due to its relatively small size and restricted exchange with the global ocean (Giorgi, 2006). This sensitivity is expected to manifest in changes of salinity distribution, consistent with an

intensified hydrological cycle in response to global warming conditions (Held and Soden, 2006; Huntington, 2006; Durack et al., 2012; Greve et al., 2014). As a result, increased evaporation and reduced freshwater inflows in the Mediterranean region have already been observed, leading to notable changes in surface salinity (Myers and Haines, 2002; Skliris et al., 2014). These changes in salinity and water mass are not confined to the region alone—variations in the density and stratification of





Mediterranean outflow can potentially affect broader global ocean circulation (Reid, 1979; Millot et al., 2006; Calafat et al.
2012; Ivanovic et al., 2014).

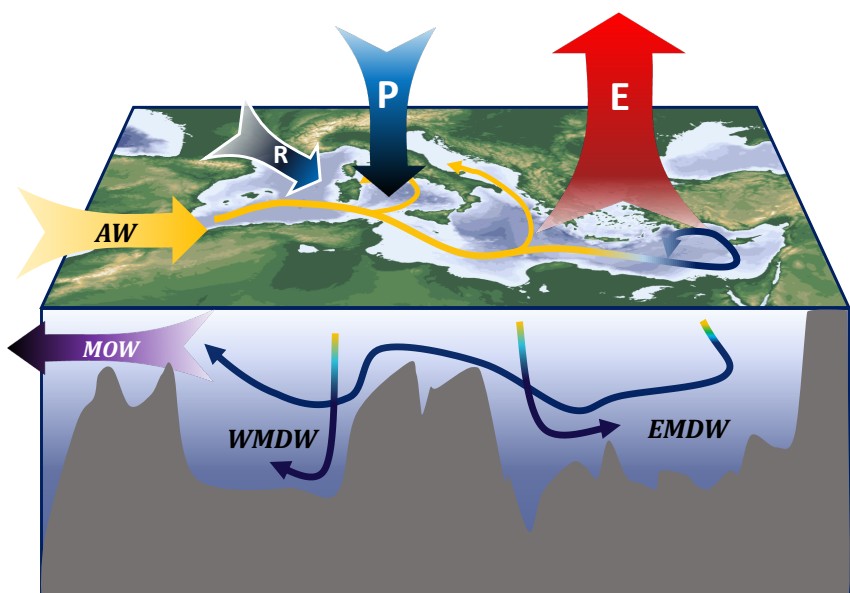

**Figure 1 Schematic illustration of the Mediterranean basin and the main budget terms at the sea surface and the Strait of Gibraltar. Red, blue, and gray arrows represent the evaporation (E), precipitation (P) and river runoff (R) at the sea surface. Yellow arrows depict the inflow of Atlantic Water (AW), and purple arrows represent the outflow of Mediterranean Outflow Water (MOW). Not all branches of the sub-basin circulation are shown. The yellow-blue curves indicate the AW circulation pathways. The Eastern Mediterranean Deep Water (EMDW) and Western Mediterranean Deep Water (WMDW) formation sites are also indicated.**

The salinity and mass variations in the Mediterranean basin are directly influenced by fluxes across the basin's boundaries (Figure 1). The two primary sources are air-sea interactions and water exchange with the North Atlantic through the Strait of Gibraltar. The freshwater flux mainly consists of evaporation (E) and precipitation (P), and drives the Mediterranean Sea
dynamics by a persistent water deficit, resulting from the imbalance between high evaporation rates and relatively low precipitation. Although river runoff (R) from major rivers contributes to the freshwater input, its impact is much smaller compared to evaporation and precipitation (Jordà et al., 2017a).

The mass loss caused by strong net evaporation in the Mediterranean Sea is replenished by inflows from the North Atlantic through the Strait of Gibraltar. The circulation at the Strait of Gibraltar can be approximated as a two-layer system, with the
upper layer carrying the relatively fresher Atlantic Water (AW) eastward, and the lower layer transporting saltier Mediterranean Outflow Water (MOW) westward at depths below 150 m. These two water masses are linked through the Mediterranean's internal thermohaline circulation and interact at sub-basin and mesoscale levels (Millot and Taupier-Letage, 2005), characterizing the overall dynamics of the Mediterranean basin (Tsimplis et al., 2008).

It is well established that the mass exchange through the Strait of Gibraltar significantly influences the Mediterranean Sea,
particularly on the regional sea-level trend (Calafat et al., 2010; Pinardi et al., 2014). Unlike the global ocean, where 50–70%





of total sea-level change is attributed to the steric component (Storto et al., 2019), the steric effect only accounts for about 20% of sea-level changes in the Mediterranean basin (Calafat et al., 2012). The exchange through the Strait is also solely responsible for changes in the basin's salt content, as surface fluxes only alter freshwater content, not actual salt.

However, the role of the air-sea freshwater fluxes is often understated despite they directly reflect the change of water cycle.
The air-sea freshwater fluxes act as both mass and volume fluxes: as a mass flux, they can be measured through ocean bottom pressure, while as a volume flux, their influence is reflected in salinity. In the open ocean, the effects of air-sea interactions are more evident over the long term (Yu, 2011; Hasson et al. 2013). But in the enclosed basins like the Caspian Sea, evaporation could dominant the sea-level trends (Chen et al., 2017), and play a major role in modulating the local salinity levels (Kara et al. 2010). In the semi-enclosed Mediterranean, these effects are often masked by the replenishment at the Strait of Gibraltar,
making it challenging to estimate how sensitive mass and salinity levels are to changes in air-sea freshwater fluxes.

While growing interest are found in using ocean salinity to quantify water cycle changes (Vinogradova and Ponte 2013, 2017; Nan et al. 2015), some researches also argue that air-sea freshwater fluxes should only be associated with salt transport when total mass transport of a region is zero (Tsubouchi et al., 2012; Schauer and Losch, 2019; Bladwell et al., 2021). This premise aligns well with the Mediterranean Sea where the water mass is well balanced (Fenoglio-Marc et al., 2012; Jordà et al., 2017a;
García-García et al., 2022). However, substantial uncertainties still remain, making it challenging to quantify the relative contributions of different water exchange pathways (Schroeder et al., 2012; Jordà et al., 2017b). The water mass exchange at the Strait of Gibraltar was estimated at 1±3 mm/yr for the period 2005–2010, though longer timescale averages may offer smaller uncertainties (García-García et al., 2022). The greater challenge lies in determining the salt budget. Due to limited observations, the mean salt flux through the Strait of Gibraltar over the past four decades was estimated at $-1.5\pm6.5 \times 10^6$ kg/s,
with uncertainties more than four times larger than the mean, despite an evident salinity increase in the Mediterranean region over the past decades (Jordà et al., 2017a). These fluxes at the boundaries of the Mediterranean basin also vary significantly across a wide range of spatial and temporal scales, making it more difficult to understand their variability and how they balance each other at this level (García-García et al., 2022).

In this study, we conduct an exploratory analysis using the dynamically consistent ocean state estimate produced by the
Consortium for Estimating the Circulation and Climate of the Ocean (ECCO version 4; Forget et al. 2015). This state estimate provides a robust framework for investigating the Mediterranean's mass and salinity fluxes at basin-wide scales, offering valuable insights into the region's water cycle dynamics and its connections to surface and oceanic forcing. While ECCOv4's coarse resolution poses challenges for resolving finer-scale processes, such as subbasin circulation, it serves as a valuable tool for identifying large-scale trends and establishing a foundation for future research. Our approach focuses on diagnosing the
mechanisms driving the temporal variability of mass and salinity budgets and understanding the links between the Mediterranean's exchanges with the atmosphere and the broader North Atlantic system.





Data and methods are described in Section 2. Section 3 presents the analysis of mass and salinity variability for the entire Mediterranean Sea, based on budget diagnostics with ECCOv4. In Section 4, we discuss model resolution, uncertainties, and potential implications to guide further detailed investigations. The study's conclusions are summarized in Section 5.

## 2 Data & Methods

### 2.1 ECCO Estimate and Its Evaluation

ECCO version 4 release 4 (v4r4) is an ocean state estimate that integrates the Massachusetts Institute of Technology General Circulation Model (MITgcm) (Marshall et al. 1997; Adcroft et al. 2004) with a wide array of observational data (Forget et al. 2015; ECCO Consortium et al. 2021). In this framework, observations are assimilated using an optimized least-squares method, ensuring dynamic and kinematic consistency without artificial heat or buoyancy sources (Heimbach et al. 2005; Wunsch et al. 2009; Wunsch and Heimbach, 2013). Argo temperature and salinity profiles and other CTD profiles from world ocean Database are used to constrain the ECCO v4r4 solution (Fukumori et al., 2017). At the sea surface, ECCO is constrained by forcing derived from the ERA-Interim reanalysis dataset (Dee et al., 2011). ECCO v4r4 spans the period 1992–2017, with a global domain and 50 vertical layers. Its resolution is 1° zonally and varies meridionally, from 1/3° at the equator to 1° at midlatitudes. For this study, we focus on the period from 2003 to 2017, during which the quality of observational data is significantly improved, enabling robust analysis of salinity, surface freshwater flux, and bottom pressure changes.

Although many previous studies have demonstrated that ECCO estimates reliably represent in situ measurements for salinity, sea level, and other variables (e.g., Stammer et al., 2004; Wunsch and Heimbach, 2007; Liu et al., 2019), this study seeks to further reduce uncertainties specifically associated with the Mediterranean Sea region. The salinity field is from the UK Met Office Hadley Centre EN4.2.2 (Good et al 2013). EN4 is an objectively-analyzed monthly dataset, covering the period from 1950 to 2016, with a horizontal resolution of 1 degree and a vertical resolution of 42 levels. It has been reported that EN4 displays some spurious salty bias after 2015 (Ponte et al., 2021; Liu et al., 2024). However, it has not been observed in the Mediterranean area (Liu et al., 2020).

The precipitation product used is the latest version, 2.3, from the Global Precipitation Climatology Project (Huffman et al. 2009). The GPCP product is created by combining various satellite and gauge-based datasets to form a coherent spatial and temporal representation. The evaporation data is obtained from the OAFlux (Yu et al. 2007), which is generated using an objective analysis method that integrates satellite and atmospheric reanalysis output, and calculates global surface fluxes using the state-of-the-art bulk flux parameterizations.

The study period (2003–2017) was selected to align with the onset of widespread Argo float deployments, which largely enhanced observational coverage and data quality in the Mediterranean Sea. While Argo data are not directly utilized in this study, they underpin the ECCO solution and contribute to the EN4 dataset, ensuring greater reliability and consistency in the input data during this period.





**Figure 2 Timeseries and time-mean spatial patterns of freshwater flux and mean salinity (0–150 m) in the Mediterranean Sea,**
**comparing ECCO outputs with other datasets. All data are interpolated onto the ECCO grid. The reference freshwater flux is**
**derived from GPCP and OAFlux, while salinity data are sourced from EN4.**

The area-averaged timeseries from these datasets were compared with the monthly mean ECCO anomaly (Figure 2). Overall,
the surface freshwater flux term of ECCO strong align with the observational *P-E*, with a correlation of 0.92 for surface
freshwater flux and R² (the proportion of variance explained by the seasonal cycle) of 0.72. Discrepancies are observed,
particularly during the winter months, where ECCO values are lower by 0.01–0.02 Sv compared to observational-based
estimates. This discrepancy could come from the river runoff term *R*, which is incorporated into the total freshwater flux. In
ECCO, river runoff is derived from observed seasonal climatology, applied as a mass flux over several surface grid cells near



river mouths (Fekete et al., 2002; Stammer et al., 2004; Feng et al., 2021). Additionally, ECCO's freshwater flux estimates may also be influenced by precipitation biases in the ERA-Interim (Turuncoglu, 2015; Grist et al., 2016).

For salinity, ECCO shows good agreement with EN4, with a correlation of 0.70 and similar long-term trends of approximately 0.01 per year. However, differences arise at seasonal and shorter timescales. ECCO's seasonal cycle has an $R^2$ of 52%, which is substantially higher than EN4's (less than 20%).

The spatial patterns of the time-mean values are also compared. Overall, the time-mean patterns from ECCO align reasonably well with the observational data products. The correlation for freshwater flux and salinity patterns ranges between 0.6 and 0.7

($p < 0.01$), respectively. Overall, ECCO captures the primary features of the observed oceanic variables in the Mediterranean Sea with reasonable accuracy (Fukumori et al. 2007; García-García et al. 2010; Soto-Navarro et al., 2010; Calafat et al. 2012). This gives us confidence in using ECCO for salinity and mass budget analyses in the region.

**2.2 The Calculation of Salinity and Mass Budgets**

Below we provide a brief summary of the salinity and mass budget analyses with the ECCO estimates. In general, our approach

is very consistent with other budget analysis using ECCO (e.g., Tesdal & Abernathey 2021; Siddiqui et al. 2024), which benefit greatly from the provided diagnostic terms. Details on how to close the budgets are provided in Forget et al. (2015) and Piecuch (2017).

In the Mediterranean Sea, significant spatial differences are observed at the sub-basin scale (Bonaduce et al., 2016; Mohamed et al., 2019). This spatial variability is caused by the complex thermohaline changes and local circulation (Menna et al., 2012;

Mauri et al., 2019; Menna et al., 2019; Poulain et al., 2021). Compared to other regional models (Escudier et al., 2021; Meli et al., 2023), ECCOv4r4's coarse resolution may limit its ability to confidently resolve these sub-basin and mesoscale processes, and it is unclear how precise the flux estimates must be to ensure accurate simulations and avoid potential model drifts. Therefore, in this study, we focus on the Mediterranean basin as a whole, rather than attempting to resolve the sub-basin processes in detail. More discussion on the model resolution and uncertainty will be discussed in later section.

The budgets for the whole Mediterranean Basin can be described as a simple box model. The balance in the Mediterranean Sea budgets is between two boundary terms, the surface flux and the inflow through the Strait of Gibraltar. Since ECCO uses volume-conserving Boussinesq approximation (Greatbatch 1994; Marshall et al. 1997), at each grid, the equation for mass budget of the whole water column can be estimated as:

$$\rho_{sw} \frac{\partial P_b}{\partial t} \approx -\nabla(\rho_{sw}\boldsymbol{u}) + \rho_{fw}F_{fw} \qquad (1)$$

where $P_b$ is the bottom pressure equivalent water thickness, $\boldsymbol{u}$ is the horizontal velocity, $F_{fw}$ is the surface freshwater flux ($P+R-E$). For the benefit of discussion, we do not separate the individual sources of freshwater inputs (i.e., $P+R$ are treated as a single term). $\rho_{fw}$ and $\rho_{sw}$ are the freshwater and seawater density, respectively.



The left-hand side represents the tendency term of bottom pressure, i.e., the rate of OBP change, the first right-hand side term is the convergence of seawater, and the second term marks the contribution of surface freshwater flux.

Given the Mediterranean Sea's semi-enclosed nature, when integrating equation 1 over the entire basin, the integral of the convergence term naturally equals to the net influx through the strait, and the integral of the second term represents the total freshwater flux at the sea surface.

At the Strait of Gibraltar, we divided the water column into two layers to estimate the inflow and outflow: the upper layer (0–150 m) represents the transport of the AW, while the lower layer (150 m to bottom, ~300 m) represents the MOW.

The salinity budget in the Mediterranean Sea is balanced in a similar form. The local salinity conservation can be simplified as:

$$\frac{\partial S}{\partial t} = -\nabla(S\boldsymbol{u}) + D_s + F_{fw}\,\tilde{S} \qquad (2)$$

where $\tilde{S}$ is the local surface salinity, and $D_S$ represents the subgrid-scale processes parameterized as mixing (diffusive salt flux). The sum of the first two terms on the right-hand side describes the total flux of oceanic transport, which, when integrated over
the entire basin, corresponds to the net influx of salinity through the strait.

It is very important to emphasize that equation 2 is constructed for salinity budget, not for salt budget. That means the dilution/concentration of salinity due to the freshwater exchange is represented explicitly in ECCO as a virtual salt flux using the local surface salinity (Forget et al., 2015). This is because the grid-cell thickness in MITgcm is held fixed with linear free-surface method, where column thickness and grid-cell thickness are fixed in time. Similarly, the first term on the right-hand
side, which represents the advection of salinity, includes two distinct physical processes that contribute to salinity changes (Piecuch, 2017): one process represents the overall dilution/concentration due to the convergence/divergence of the mass transport ($S\nabla\boldsymbol{u}$), while the other reflects the exchange of salt content carried by the advective flow ($\boldsymbol{u}\nabla S$).

The focus of this study is on the interannual variability of the fluxes. The non-seasonal signal is obtained subtracting the climatology from the original timeseries instead of a fitted annual sinusoid, since the annual variation is not always sinusoidal
in shape (García-García et al. 2022). This approach is consistent with previous studies using ECCO to investigate Mediterranean Sea variability and phenomena associated with Gibraltar Strait transport (Fukumori et al., 2007; Menemenlis et al., 2007; Volkov and Landerer, 2015). These studies highlight a strong correlation between OBP and sea level on non-seasonal timescales, providing a more accurate representation of the relationship between mass (OBP) variations and volume (sea level) variations. By our estimation, the seasonal cycle accounts for less than 10% of the total variance in the mass flux,
while the salinity component contributes slightly more, below 30%.

Due to the dual nature of air-sea fluxes as both mass and volume fluxes, we represent the mass flux in Sverdrups (Sv) for consistency. Since salinity is a unitless measure, we also express salinity flux in Sv in comparing both types of fluxes.





## 3 Results

### 3.1 Non-seasonal Variability of Mass and Salinity Fluxes

In this section we present the estimates for each flux term of Eq. 1 and Eq. 2 applied to the entire Mediterranean Sea. Figure 4a presents the temporal variability of the Mediterranean Sea's mass budget from 2003 to the end of 2017. In general, the net influx at the Strait of Gibraltar (i.e., inflow plus outflow, labeled "*Strait*"), shows large month-to-month variability: its standard deviation is 2.4 times larger than the surface freshwater flux (labeled "*Surface*").

The mass tendency term, i.e., the sum of *Strait* and *Surface* terms, is labeled "*Total*" and mainly driven by the net influx at the
Strait. The Pearson correlation coefficient between *Strait* and *Total* terms' timeseries is 0.96 ($p < 0.01$), while there is no significant correlation between *Total* and *Surface* terms ($r < 0.2$). The coefficient of determination ($R^2$) of *Strait* to *Total* term is 0.92, showing that the net influx explains nearly all of the variance in the total mass tendency. Previous studies have shown that the mass exchange through Gibraltar significantly impacts the Mediterranean, dominating the mean sea-level trend in the region (Calafat et al., 2010; Pinardi et al., 2014).

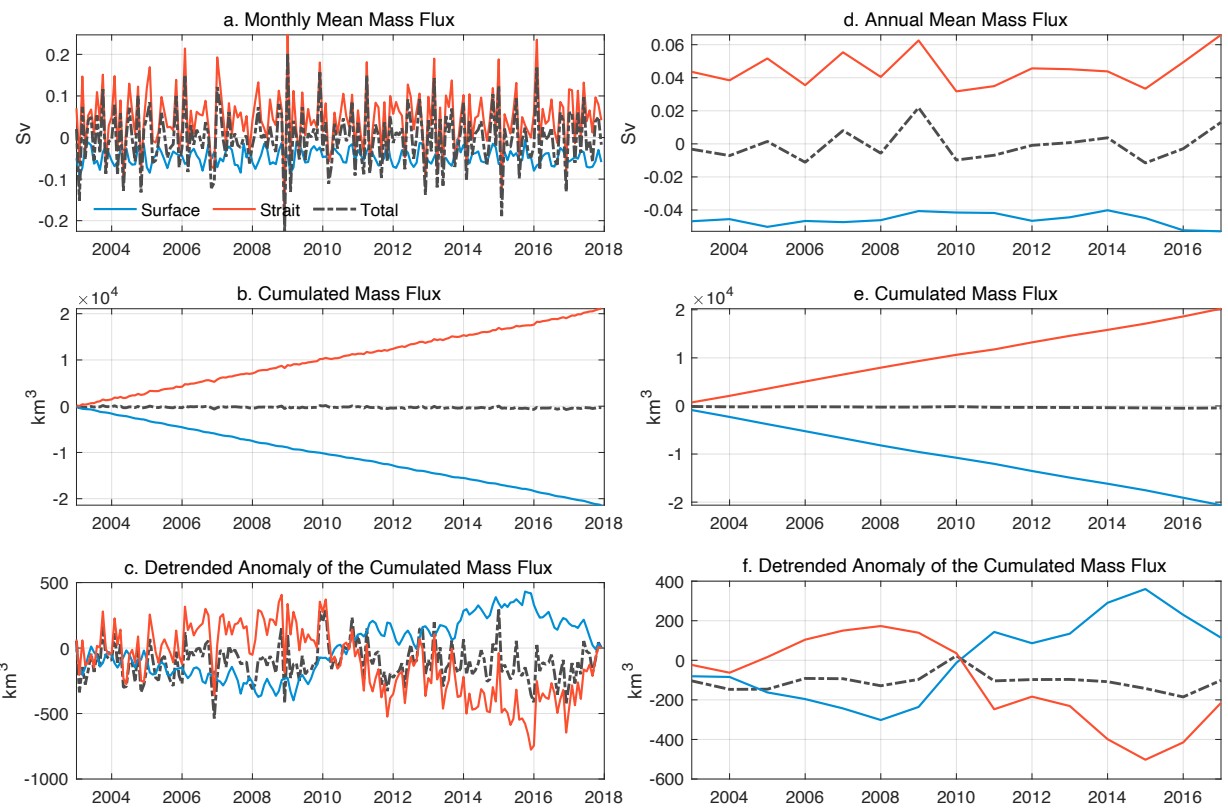

**Figure 3 Mass fluxes in the Mediterranean Sea. (a) Monthly mean timeseries of Mass flux through the sea surface, Strait of Gibraltar, and the sum of both; (b) Temporally cumulated mass fluxes; (c) Same as b but detrended. (d-f) Same as a-c but with annual means.**





The interannual variability of both the *Strait* and *Total* terms is evidently much smaller on a year-to-year scale compared to month-to-month fluctuations (Figure 3d). The transport through the Strait of Gibraltar exceeded 0.06 Sv in 2009 and 2017,

while the lowest values occurred in 2010 and 2015, around 0.03 Sv. In contrast, the *Surface* term exhibits very limited interannual variability, with a range of only 0.01 Sv, reaching a maximum in 2014. Similar numbers have been previously reported in the literature; García-García et al. (2022) provided annual mean estimates of net exchange at the Strait of Gibraltar at 0.04 Sv in 2008, and 0.0224 Sv in 2010.

Linear trends of the fluxes were also estimated, showing a small upward trend in both the *Strait* and *Total* terms ($O(10^{-4}$ Sv

205 per year)). However, these trends are not statistically significant as their 95% confidence intervals include zero. This indicates that the net transport through the Strait of Gibraltar, as well as the overall mass within the Mediterranean Sea, remained highly stable over the study period. On average, the total mass tendency is close to zero ($O(10^{-4}$ Sv)), and the net oceanic influx through the Strait of Gibraltar and the surface freshwater flux appear to be balanced almost simultaneously, with no noticeable lag.

We then examined the cumulative mass flux timeseries, which represents the contribution of each flux term to the overall mass gain/loss over time (Figure 3b). The surface flux contribution shows a consistent downward slope at a rate of -1390±18 km³/year, reaching a cumulative mass loss of approximately 20,000 km³ over the 15-year period. In contrast, the net inflow through the Strait of Gibraltar contributes a mass gain at a slightly smaller rate of approximately 1368±20 km³/year. Overall, the total water mass in the Mediterranean remains relatively stable from 2003 to 2017, with a very slight and insignificant

decreasing trend of -22±24 km³/year, suggesting a near-equilibrium state.

To better observe the interannual variability in the contributions of mass fluxes, we further detrended the cumulated mass flux timeseries (Figure 3c). magnitude of the detrended timeseries are approximately 2 orders smaller than the original cumulated timeseries. The results reveal that the *Surface* and *Strait* terms exhibit nearly exact opposite variations, although the *Surface* term has relatively smaller month-to-month fluctuations. This opposing pattern is expected, given that the total mass in the

220 Mediterranean Sea remains nearly unchanged (as seen in Figure 3b).

A notable turning point occurs in the winter of 2010/2011. Prior to this, from 2003 to 2010, the surface flux shows a predominantly negative anomaly, followed by a subsequent seven-year period of positive anomalies. The *Strait* term exhibits the exact opposite pattern, with positive anomalies before 2010 and negative anomalies afterward. Notably, both fluxes tend to contribute to the removal of freshwater from the Mediterranean basin. Before 2010, the freshwater loss from surface fluxes

exceeded the net inflow through the strait by approximately 100 km³. After 2010, the *Strait* term has a negative anomaly and reaches values as low as -500 km³, showing a larger deficit. The connection between this interannual variability and regional climate index, namely the North Atlantic Oscillation (NAO), will be examined later.




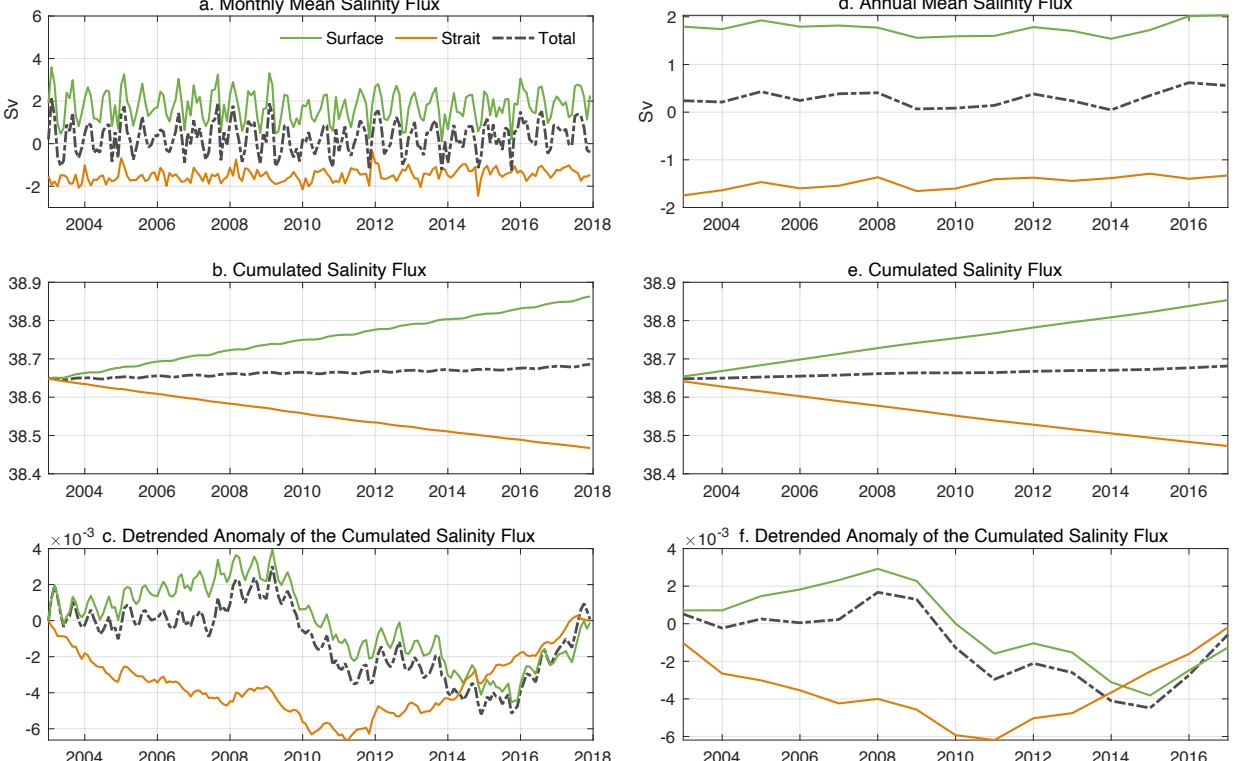

**Figure 4 Salinity fluxes in the Mediterranean Sea. (a) Monthly mean timeseries of salinity flux through the sea surface, Strait of**
230 **Gibraltar, and the sum of both; (b) Temporally cumulated salinity fluxes expressed as the equivalent of mean salinity change in the**
**Mediterranean Sea; (c) Same as b but detrended. (d-f) Same as a-c but with annual means.**

Figure 4 presents the salinity fluxes for the Mediterranean Sea. Unlike the mass budget, the *Surface* term exhibits considerably

larger month-to-month variability in salinity, with a standard deviation 2.5 times greater than that of the *Strait* term. Moreover,

the surface freshwater flux demonstrates a strong linear relationship with the total salinity tendency: the correlation between

235 surface flux and *Total* salinity tendency is exceptionally high, exceeding 0.97 (p < 0.01) with an R² value of 0.95, indicating

that air-sea fluxes alone account for nearly all observed salinity changes. In contrast, the net influx through the Strait of

Gibraltar explains only 4% of the salinity variability, with a much weaker correlation of 0.25. This is because freshwater,

which contains no salt, could substantially dilutes the denser, saltier Mediterranean waters when introduced, impacting the

overall salinity dynamics within the basin.

Regarding the linear trends, a significant increase was found in the *Strait* term at 0.02±0.01 Sv per year, while no significant

trend was observed in the *Surface* and *Total* terms. Since there is no substantial long-term change in the overall mass transport

through the Strait of Gibraltar (as in Figure 3), this is likely because the inflowing North Atlantic water is becoming saltier

over time, which is consistent with some recent findings (Bates & Johnson 2020; Sukhonos et al. 2024).



In Figure 4b, we present the cumulative contribution of salinity fluxes to the mean salinity level in the Mediterranean Sea.
Over the study period, the mean salinity of Mediterranean seawater shows a steady increase at a rate of approximately $2.2\pm0.2 \times 10^{-3}$ per year, translating to a total salinity increase of about 0.03 over 15 years. The air-sea freshwater flux, driven primarily by substantial net evaporation, contributes significantly to this trend. On its own, it would have raised the mean salinity by 0.2 over the period, with a rate of $14.0\pm0.2 \times 10^{-3}$ per year. In comparison, the contribution from the inflow through the Strait of Gibraltar is estimated at $-12.1\pm0.2 \times 10^{-3}$ per year, accumulating to a reduction of 0.17 over 15 years.

The detrended cumulative salinity timeseries are presented in Figure 4c. The *Surface* and *Total* terms show a very high correlation coefficient of 0.95 ($p < 0.01$) and $R^2$ (0.88), showing the dominant influence of air-sea interactions on salinity variations. Prior to 2010, both terms exhibit positive salinity anomalies, while negative anomalies are prevalent in the years that follow. In contrast, the Strait term maintains a persistent negative anomaly throughout the period, reaching its lowest point in 2011. It is also out of phase with the other flux terms, highlighting its distinct behavior relative to surface-driven salinity
changes.

## 3.2  Influence of NAO on Salinity and Water Mass Transport

The NAO is one of the primary climate modes affecting Europe and the Mediterranean region, with a significant impact on both weather and oceanic circulation (e.g., Hurrell, 1995; Vigo et al., 2011). The NAO is known to influence large-scale sea level pressure patterns over the Mediterranean Sea. Particularly relevant to this study, it modulates wind patterns near the Strait
of Gibraltar, affecting non-seasonal water mass transport and variations in Mediterranean sea levels (Fukumori et al., 2007; Menemenlis et al., 2007; Landerer and Volkov, 2013; Piecuch and Ponte, 2014). Importantly, the NAO influences the Mediterranean region across multiple timescales, from seasonal to interannual and decadal (Gomis et al., 2008; Tsimplis et al., 2008; Calafat et al., 2010; Calafat et al., 2012). These processes offer a framework for understanding the relationship between the NAO and the interannual variability of the observed timeseries of mass and salinity fluxes in the Mediterranean.



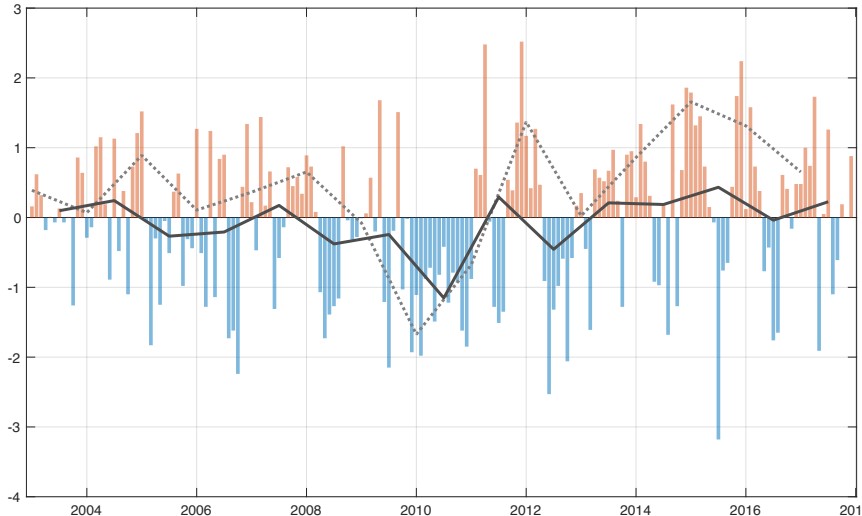

**Figure 5 The standardized monthly NAO index with the annual mean (black solid line) and the winter mean (gray dotted line). Winter means are calculated from December to February. Red bars represent months with positive NAO values, while blue bars represent months with negative values.**

The monthly mean NAO index, obtained from NOAA (https://www.cpc.ncep.noaa.gov/products/precip/CWlink/pna/nao.shtml), is displayed in Figure 5. The index reveals that the NAO remained in a positive-to-neutral phase from approximately 2003 to 2008, shifted to a negative phase from 2008 to 2011, and then returned to a positive phase afterward. We also calculated annual and winter seasonal averages for the NAO timeseries, as previous studies have shown that the NAO is particularly influential on Mediterranean weather patterns during the winter season (e.g., Castro-Díez et al., 2002).

Mariotti et al. (2002) found significant correlations between the NAO and both annual and winter averages of precipitation and net precipitation across the broader Mediterranean region. In our study, we also observed a significant correlation between the winter NAO and the air-sea mass flux shown in Figure 3d (blue), with a correlation coefficient of -0.51 ($p < 0.05$). Regarding salinity fluxes in Figure 4d, all annual fluxes show significant correlations with the winter NAO. Both the Surface and Strait terms are correlated with the winter NAO between 0.5 and 0.6 ($p < 0.05$), while the *Total* term, i.e., the sum of the *Surface* and *Strait* terms, correlates at 0.62 ($p < 0.01$).

When fluxes are cumulated over time, the phases of the timeseries naturally shift. For the timespan discussed here, this results in a shift in phase from a cosine-like pattern to a sinusoidal form, as illustrated in Figures 2d and 2f. Notably, the cumulative salinity flux of the *Strait* term shows a moderate correlation with the NAO index, with a correlation coefficient of 0.68 ($p < 0.05$).



## 3.3 Roles of Air-Sea Fluxes and Oceanic Exchange During 2003-2017

This section provides a summary of the estimated mass and salinity budgets, broken down by their components as represented in Eq.1 and Eq.2, along with associated uncertainties for the Mediterranean Sea over the period 2003–2017. Since these estimates are derived using ECCO, the budget calculations are inherently balanced. The budget diagram shown in Figure 6 is on the mass balance (top) and the salinity imbalance within the Mediterranean basin (bottom). Over this 15-year period, the long-term mean water mass change is calculated at 0.001±0.018 Sv, while the salinity change is estimated at 0.29±0.09 Sv. This reflects a balanced state in terms of mass, contrasted with a slight imbalance in salinity. These differences arise from the distinct roles of surface processes, which exchange freshwater, and oceanic exchange of saltwater through the Strait of Gibraltar.

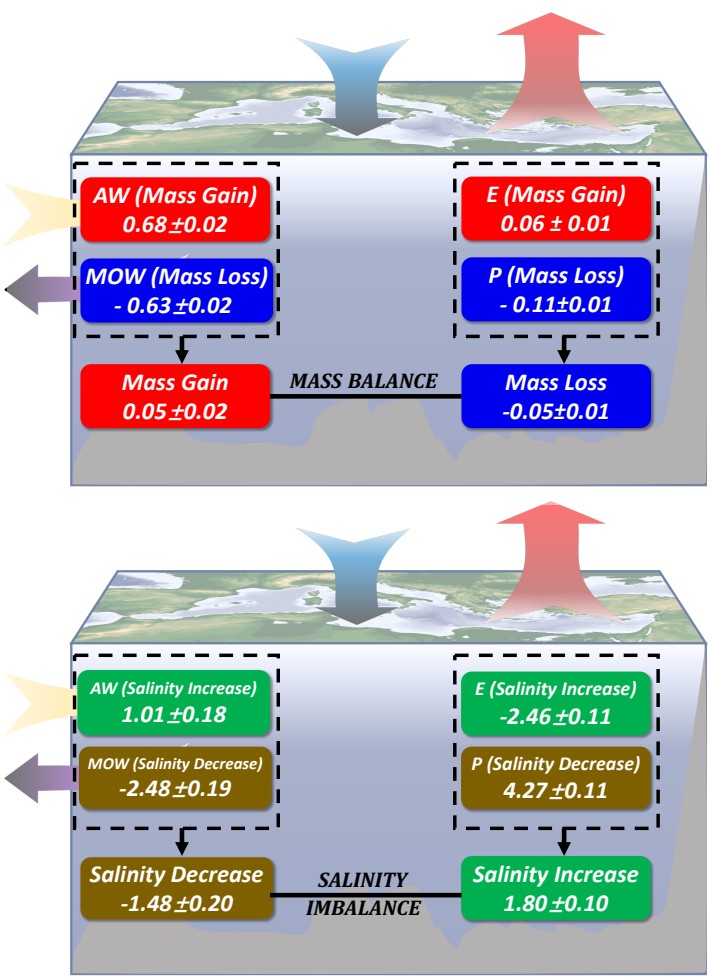

**Figure 6 15-year mean mass (top) and salinity (bottom) budget for the Mediterranean Sea. Red and blue boxes mark the mass gain and mass loss, brown and green boxes mark the salinity increase and decrease, respectively. In each panel, the contributions from air-sea fluxes are on the right, and the exchange through the Strait are on the left. The uncertainties are calculated as the standard**





error of the means, σ/√n, where σ is the standard deviation of the corresponding term and n = 15 for non-seasonal fluxes. Units are Sv.

**Table 1 Estimates of the different components of the Mediterranean Sea mass and salinity with uncertainties for the period 2003–2017. Note that the salinity change is separated into contributions from changes in salt content, and changes in salinity due to dilution/concentration effect.**

| | Surface Fluxes | | | Strait of Gibraltar | | |
|---|---|---|---|---|---|---|
| Processes | *E* | *P* | *Total* | *AW* | *MOW* | *Total* |
| **Total Mass Change** | **-0.11±0.01** | **0.06±0.01** | **-0.05±0.01** | **0.68±0.02** | **-0.63±0.02** | **0.05±0.02** |
| Salt Content Change | | | | 25.21±0.18 | -24.91±0.19 | 0.30±0.20 |
| Dilution/Concentration | 4.27±0.11 | -2.46±0.11 | 1.80±0.10 | -24.21±0.15 | 22.43±0.16 | -1.78±0.17 |
| **Total Salinity Change** | **4.27±0.11** | **-2.46±0.11** | **1.80±0.10** | **1.01±0.18** | **-2.48±0.19** | **-1.48±0.20** |

Starting with the mass budget, which is relatively straightforward: as outlined in the Data & Methods section, at each interface (air-sea and the Strait of Gibraltar) we can break the fluxes down into two primary processes. At the surface, these processes are evaporation and precipitation (including runoff). Evaporation results in a mass loss of -0.11±0.01 Sv, while precipitation adds a mass gain of 0.06±0.01 Sv. This leads to a net mass loss of -0.05±0.01 Sv from all air-sea freshwater fluxes.

At the Strait of Gibraltar, the two processes are the inflow of AW and the outflow of MOW. The inflow through the upper

layer brings in 0.68±0.02 Sv of AW, while the outflow in the lower layer exports -0.63±0.02 Sv of MOW. The net result is a modest positive mass gain, on the order of one magnitude smaller than either of the individual flows. Overall, the Mediterranean Sea gains water mass through net inflow at an average rate of 0.05±0.02 Sv, which closely aligns with values reported in the literature. For instance, Jordà et al. (2017a) derived a similar number at 0.065±0.033 Sv from nearly 20 independent observational estimates.

Regarding the salinity budget, as noted previously, salt and salinity transport are only meaningful for understanding freshwater fluxes when the total mass transport within a closed region is zero (Tsubouchi et al., 2012; Schauer and Losch, 2019). This is an ideal condition in our case, as we have already established a mass balance within the semi-enclosed Mediterranean basin.



In ECCO, evaporation and precipitation processes are introduced as virtual salt fluxes for they do not introduce any actual salt, but they can still alter salinity levels by changing the total volume of the water body. Evaporation increases the salinity level

by removing freshwater, resulting in a concentration effect equivalent to 4.27±0.11 Sv. On the other hand, precipitation dilutes the seawater, leading to a salinity reduction of -2.46±0.11 Sv. Together, these air-sea fluxes contribute a net positive of 1.80±0.10 Sv.

In contrast to the surface processes, the salinity exchange through the Strait of Gibraltar operates in a more complex manner. As outlined in the methods section, salinity flux through the strait can be split into two main components: one associated with

the transport of salt content due to mass flow ($\boldsymbol{u}\nabla S$), and another due to changes in volume and mass, which result in salinification or dilution effects ($S\nabla\boldsymbol{u}$). Starting with the salt content transport $\boldsymbol{u}\nabla S$, ECCO estimate a net salt flux into the Mediterranean through this exchange at 0.30±0.20 Sv. This is driven by the exchange between the AW and the relatively saltier MOW at the Strait of Gibraltar. Specifically, the AW brings in 25.21±0.18 Sv of salt content, while the MOW carries out -24.91±0.19 Sv.

We then consider the dilution and concentration effects resulting from the introduction of AW and MOW at the Strait of Gibraltar, which is driven by the density differences between AW, MOW, and the average Mediterranean seawater. In this context, assuming the total salt content within the Mediterranean Sea remains constant, the addition of AW (with relatively lower salinity) and the removal of MOW (with higher salinity) lead to a net dilution effect due to changes in volume and density; specifically, the water exchange at the Strait of Gibraltar would reduce the salinity by approximately -1.78±0.03 Sv

due to a net dilution effect. After the adjustment, the net influence of the water exchange at the Strait of Gibraltar shifts from a net gain in salt content to an overall reduction in salinity, estimated at -1.48±0.20 Sv. This hilights the critical role of volume/density changes in the Mediterranean Sea's salinity budget, which significantly impact the balance of salinity in this semi-enclosed basin.

## 4 Discussion of ECCO Resolution and Uncertainty

The use of ECCO v4r4 in this study provides valuable insights into the Mediterranean Sea's salinity and freshwater budgets. However, the model's coarse 1° resolution limits its ability to resolve mesoscale and submesoscale processes, particularly in critical areas like the Gibraltar Strait, where fine-scale dynamics play a significant role in water and salt fluxes (e.g., Soto-Navarro et al., 2010). This limitation introduces uncertainty in capturing key aspects of the seasonal and interannual variability of freshwater and salinity trends.

As such, our analysis focuses on basin-scale budgets rather than spatial patterns of trends or decadal means, recognizing ECCO's reduced capacity to accurately represent localized circulation and exchange processes. Some recent regional ocean models, such as the Med Sea Physics reanalysis (Escudier et al., 2021) with 1/24° resolution, are better equipped to capture these dynamics, including contributions from smaller branches like the Black Sea inflow (Potiris et al., 2024; Mamoutos et al.,





2024), which ECCO may underrepresent due to its resolution constraints. ECCO's low resolution also makes it challenging to
accurately capture vertical variations in salinity, particularly in the complex intermediate and deep water masses of the
Mediterranean. A further direct comparison between ECCO and higher-resolution datasets would help quantify the
uncertainties associated with resolution differences and refine our understanding of regional and basin-wide trends.

Given the complexity of oceanic exchanges and salinity dynamics in the Mediterranean, future work should include a detailed
uncertainty analysis to quantify the impact of ECCO's resolution on salinity and freshwater trends. For example, large
discrepancies exist between reanalysis products in representing Mediterranean air-sea flux variations (Josey et al., 2011; Skliris
et al., 2024). Incorporating higher-resolution datasets alongside ECCO could provide a more robust framework for evaluating
these discrepancies and better understand the reliability of modeled outputs, particularly when extrapolating trends across
different temporal and spatial scales.

While this study provides a valuable starting point, we recognize the need for further efforts to address these uncertainties,
especially in validating our findings with more precise data sources. Future steps should prioritize high-resolution comparisons
and improved validation to advance our understanding of the Mediterranean's salinity and freshwater budgets.

## 5 Conclusion

This study provides a comprehensive assessment of the mass and salinity budgets in the Mediterranean Sea from 2003 to 2017
using the ECCO state estimate. By focusing on the contributions of two primary boundary fluxes—surface freshwater fluxes
and the saltwater exchange at the Strait of Gibraltar—we reveal the complex dynamics that govern the Mediterranean's mass
and salinity variations. Our analysis highlights the critical role of air-sea freshwater fluxes, which dominate long-term balance
and variability as both mass and volume fluxes.

The surface fluxes remove water mass from the basin through net evaporation, resulting in an increase in salinity levels by
approximately $1.80 \pm 0.10$ Sv and accounting for most of the observed salinity variability. In comparison, the Gibraltar
exchange replenishes this mass loss and contributes approximately $0.30 \pm 0.20$ Sv of salt. However, due to the density
differences between AW, MOW, and Mediterranean water, there is a net salinity reduction of $-1.48 \pm 0.20$ Sv through the
strait. Over the study period, these dynamics collectively lead to a salinity increase of $0.29 \pm 0.09$ Sv. These diagnosed numbers
are largely consistent with values reported in previous studies (Calafat et al. 2010, 2012; Jordá et al., 2017b; Soto-Navarro et
al., 2010; Llases et al., 2018).

Recent researches suggest that focusing on the salt budget, rather than the salinity budget, can improve accuracy by eliminating
dilution and concentration effects from freshwater fluxes (Schauer & Losch, 2019). This finding validates the salt budget
framework as an effective tool for disentangling the relative contributions of key processes to salinity variability.

Our results also demonstrate that the salinity increase observed in the Mediterranean basin (Malanotte-Rizzoli et al., 2014) is a steady and sustained process driven by long-term factors, such as the gradual rise in surface net evaporation and the influx

of saltier waters from the Atlantic, rather than isolated, short-term events. While these findings highlight the limitations of ECCO's coarse resolution in resolving fine-scale processes, including the complex exchange at Gibraltar, they provide a robust baseline for understanding large-scale trends. Future research, leveraging higher-resolution datasets, could refine these estimates and better capture the nuances of Mediterranean exchange dynamics.

Beyond salinity, our findings contribute to a broader understanding of sea level dynamics in the Mediterranean. Previous

studies (Fukumori and Wang, 2013; Hamlington et al., 2020) have linked regional sea level variations to changes in evaporation and precipitation, suggesting a connection to global mean sea level. By clarifying the role of air-sea fluxes in regulating water mass and salinity, we illustrate how these processes drive regional steric (halosteric) adjustments and contribute to global ocean mass variation. While temperature-driven changes remain the dominant force behind global sea level rise, our study underscores the critical, yet often understated, influence of air-sea fluxes in shaping regional and global trends.

In summary, this study reinforces the importance of air-sea interactions and strait exchanges in controlling the Mediterranean's salinity and mass budgets. The integration of salt and mass budgets provides a refined framework for interpreting the region's response to climatic forces, while also emphasizing the necessity of addressing model limitations. This foundational work not only bridges regional processes and global outcomes but also identifies clear paths forward for future research to address uncertainties and advance our understanding of the Mediterranean's complex dynamics in a changing climate.

**Data Availability**

The gridded EN4 used in this work is accessible at http://hadobs.metoffice.com/en4. The OAFlux is accessible at https://oaflux.whoi.edu/, and the GPCP monthly analysis is available at https://psl.noaa.gov/data/gridded/data.gpcp.html. The ECCO ocean state estimate is accessible at https://ecco-group.org/.

**Author contributions**

All coauthors defined the research problem and the conceptualization of the study. CL carried out the data analysis and produced the figures and first draft under the supervision of XL. All coauthors discussed the analysis and contributed to the writing of the final paper.

**Competing interests**

The contact author has declared that none of the authors has any competing interests.



**Financial support**

This study was supported by NASA through grant 80NSSC22K0996.

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
