# Peer review of "Salinity Trends and Mass Balances in the Mediterranean Sea: Revisit the Role of Air-Sea Freshwater Fluxes and Oceanic Exchange"

_EGUsphere, 2025_

## Community Comment (CC1)

**Literature review**

Jari Miglio

**Abstract**

*This work was conducted as part of a literature review for the course "Physics of the Hydrosphere and Cryosphere" at the Università degli Studi di Milano. Its primary objective was to summarize the key findings of the referenced article and analyze the methodology employed. During the review, two typographical errors were identified. The first appears in Section "3.1 Non-seasonal Variability of Mass and Salinity Fluxes", where Figure 3a (instead of Figure 4a) correctly presents the temporal variability of the Mediterranean Sea's mass budget. The second error is in Section "3.3 Roles of Air-Sea Fluxes and Oceanic Exchange During 2003–2017", where, in Figure 6, the labels for evaporation and precipitation are inverted: evaporation, which represents mass loss, should correspond to the blue box, while precipitation, which represents mass gain, should correspond to the red box. Additionally, Section "2.2 The Calculation of Salinity and Mass Budget" requires further clarification, as the equations proposed for the mass and salinity balances in the Mediterranean basin are dimensionally inconsistent. A discussion of this issue is provided in the corresponding section of this review. All content in this literature review has been revised and approved by Professor Mauro Giudici.*

**1    Introduction**

The Mediterranean Sea is known to be sensitive to climate change due to its relatively small size and restricted exchange with the global ocean. Global warming has led to significant alterations in key physical quantities, such as surface salinity, which serve as indicators of the state of the Mediterranean's water masses. In particular, the overall increase in mean temperature has intensified the hydrological cycle, resulting in an increased evaporation rate and a reduction in freshwater inflows to the Mediterranean region. These changes have led to notable modifications in surface salinity.

The Mediterranean basin exhibits complex dynamics, with salinity and mass variations directly influenced by boundary fluxes, primarily air-sea interactions and water exchange with the North Atlantic through the Strait of Gibraltar. These two primary processes contribute to the basin's water balance in distinct ways:

1. Air-sea freshwater fluxes

   The freshwater flux is primarily composed of evaporation (E) and precipitation (P), driving the Mediterranean's persistent water deficit, which arises from the imbalance between high evaporation rates and relatively low precipitation. Additionally, river runoff (R) from major rivers contributes to the freshwater input, but its effect is significantly smaller compared to evaporation and precipitation. Despite its lower contribution, runoff is included in this study for completeness.

Surface fluxes only alter the freshwater content of the basin since they do not affect the total salt content and act as both mass and volume fluxes:

(a) As a mass flux, they can be measured through ocean bottom pressure (a decrease in mass due to net evaporation results in a drop in pressure, assuming other factors remain constant).

(b) As a volume flux, their impact is reflected in salinity changes (variations in water volume affect concentration, leading to changes in salinity).

These contributions arise from the strong coupling between the atmosphere and the hydrosphere.

2. Water exchange through the Strait of Gibraltar

The circulation at the Strait of Gibraltar can be approximated as a two-layer system:

(a) The upper layer carries relatively fresher Atlantic Water (AW) eastward into the Mediterranean Sea.

(b) The lower layer transports saltier Mediterranean Outflow Water (MOW) westward, at depths below 150 m.

These two water masses are interconnected through the Mediterranean's internal thermohaline circulation. The exchange through the Strait compensates for the mass loss caused by strong net evaporation and is the primary mechanism regulating the basin's salt budget.

Changes in air-sea freshwater fluxes, which reflect climate variability, have direct effects on both mass balance and salinity levels. However, in the semi-enclosed Mediterranean, these effects are often partially masked by the compensating exchange at the Strait of Gibraltar, making it challenging to provide precise estimations.

In this study, the authors conduct an exploratory analysis using the dynamically consistent ocean state estimate produced by the Consortium for Estimating the Circulation and Climate of the Ocean (ECCO version 4). This model provides a robust framework for investigating the Mediterranean's mass and salinity budgets at a basin-wide scale. The analysis focuses on diagnosing the mechanisms driving temporal variability in mass and salinity and understanding the interactions between the Mediterranean Sea, the Atmosphere, and the broader North Atlantic system.

**2    Data and Methods**

**2.1    ECCO evaluation**

The model used in this study is an ocean state estimate, where observational data are assimilated while ensuring dynamic and kinematic consistency. Observations, such as Argo temperature and salinity profiles, are used to constrain the ECCO solution, which is obtained by integrating fundamental physical conservation laws. At the sea surface, ECCO is constrained by atmospheric forcing derived from the ERA-Interim reanalysis, a historical dataset providing a wide range of atmospheric parameters.

ECCO solutions span the period 1992–2017 and have a global domain with 50 vertical layers. The model resolution is relatively coarse, with a zonal (longitudinal) resolution of 1°, while the meridional resolution varies, ranging from 1/3° at the equator to 1° at midlatitudes. It is important to emphasize that the resolution of a model refers to the longitudinal and latitudinal spacing of the computational grid, which determines how finely the model discretizes the Earth's surface.

The study focuses on the period 2003–2017, chosen to coincide with the widespread deployment of Argo floats, which significantly improved observational coverage and data quality in the Mediterranean Sea. A comparison between ECCO model outputs and observational datasets, including time series and time-mean spatial patterns of freshwater fluxes, reveals a reasonable agreement (Figure 2 of the paper). Overall, ECCO effectively captures the primary features of oceanic variables in the Mediterranean Sea with reasonable accuracy, giving the authors confidence in its use for salinity and mass budget analysis in the region.

**2.2 Calculation of Salinity and Mass Budgets**

The Mediterranean Sea is characterized by complex thermohaline variability and local circulation patterns, which lead to significant spatial differences at the sub-basin scale. However, the relatively coarse resolution of ECCO may limit its ability to accurately resolve these finer-scale processes and consequently, the precision of sub-basin simulations remains uncertain. For this reason, this study focuses on the Mediterranean basin as a whole, rather than attempting to resolve localized sub-basin processes in detail.

The mass conservation principle governing the Mediterranean Sea's budget can be described using a simplified box model, with two primary boundary terms: surface fluxes and water exchanges through the Strait of Gibraltar. In the study, the mass budget equation for the entire water column is expressed as:

$$\rho_{sw}\frac{\partial P_b}{\partial t} \approx -\nabla \cdot (\rho_{sw}\mathbf{u}) + \rho_{fw}F_{fw} \tag{1}$$

where $P_b$ is the bottom pressure equivalent water thickness, $\mathbf{u}$ is the horizontal velocity, $F_{fw}$ is the surface freshwater flux $(P + R - E)$, $\rho_{sw}$ and $\rho_{fw}$ are respectively the seawater and freshwater density. Some remarks are needed:

1. The left-hand side term represents a measure of sea mass variability, incorporating contributions from fluctuations in atmospheric sea level pressure, dynamic topography, and variations in the mass of the fluid column between the unperturbed surface and the seafloor. These factors collectively influence Observed Bottom Pressure (OBP) variability. Furthermore, OBP variability is strongly correlated with sea surface height (SSH) variability, which is also affected by atmospheric loading and the hydrological cycle. In summary, OBP can be considered a proxy for ocean mass variability, as well as for ocean circulation changes and variations in the Earth's gravity field. To illustrate this concept, let us consider a cylindrical water column extending from the seafloor to the sea surface, with a thickness $H$ and surface area $S$. The notation used in this review differs slightly from that in the referenced article: here, $P_b$ denotes the bottom pressure, while $H$ retains the same meaning as what is called $P_b$ in the article, representing the thickness of the water column. The bottom pressure at the seafloor can then be expressed as:

$$P_b = P_a + \frac{m_{sw}g}{S} = P_a + \rho_{sw}gH. \tag{2}$$

Hence the time variation of the sea bottom pressure is:

$$\frac{\partial P_b}{\partial t} = \frac{\partial P_a}{\partial t} + \frac{\partial(\rho_{sw}gH)}{\partial t} = \frac{\partial P_a}{\partial t} + \rho_{sw}H\frac{\partial g}{\partial t} + Hg\frac{\partial \rho_{sw}}{\partial t} + \rho_{sw}g\frac{\partial H}{\partial t} \tag{3}$$

$$\frac{1}{g}\frac{\partial P_b}{\partial t} = \frac{1}{g}\frac{\partial P_a}{\partial t} + \frac{\rho_{sw}H}{g}\frac{\partial g}{\partial t} + H\frac{\partial \rho_{sw}}{\partial t} + \rho_{sw}\frac{\partial H}{\partial t} \tag{4}$$

Neglecting the time variation of atmospheric pressure and seawater density we obtain:

$$\frac{1}{g}\frac{\partial P_b}{\partial t} \approx \rho_{sw}\frac{\partial H}{\partial t} \tag{5}$$

This computation aims to clarify the first term on the left-hand side of the equation. It should be interpreted as the product of seawater density and the time variation of the ocean's thickness. Alternatively, when neglecting variations in atmospheric pressure and seawater density, it represents the tendency term of bottom pressure per unit gravity, i.e., the rate of change of OBP (Observed Bottom Pressure).

The corresponding dimensions are:

$$\left[\rho_{sw}\frac{\partial H}{\partial t}\right] = \frac{\text{kg}}{\text{m}^3}\frac{\text{m}}{\text{s}} = \frac{\text{kg}}{\text{m}^2\text{s}} \tag{6}$$

and can be interpreted as:

$$\frac{\Delta M_{\text{in/out on the surface of Mediterranean basin in }\Delta t}}{\Delta t \cdot S} \tag{7}$$

2. The first term on the right-hand side represents the convergence of seawater fluxes and accounts for the time variation of seawater density due to water exchanges at the Strait of Gibraltar. The dimensional estimation of this term can be computed as follows:

$$\left[\nabla \cdot (\rho_{sw}\mathbf{v})\right] = \frac{1}{\text{m}}\frac{\text{kg}}{\text{m}^3}\frac{\text{m}}{\text{s}} = \frac{\text{kg}}{\text{m}^3\text{s}} \tag{8}$$

Since the dimensions of this quantity are not consistent with the overall equation, equation (1) must be expressed in a formally consistent manner. To achieve this, equation (1) should either be considered in its integral form, summing up all contributions over the entire Mediterranean basin, or this term should be integrated over the entire thickness of the sea.

3. The second right-hand side term marks the contribution of surface freshwater fluxes and its dimension are perfectly coherent with the dimension of the left-hand side:

$$\left[\rho_{fw}F_{fw}\right] = \frac{\text{kg}}{\text{m}^3} \cdot \frac{\text{m}}{\text{s}} = \frac{\text{kg}}{\text{m}^2\text{s}} \tag{9}$$

Thus, this term can be interpreted as the net mass of freshwater exchanged through air-ocean interactions per unit area and per unit time, i.e., the rate of mass exchange per unit surface area:

$$\frac{\Delta M_{\text{net in-out flow due to P+R-E}}}{\Delta t \cdot S} \tag{10}$$

Integrating this term over the sea surface represents the total freshwater flux at the Mediterranean basin surface.

The article also offers a formula to calculate the salinity budget in the Mediterranean Sea, balancing in a similar form. The local salinity conservation can be simplified as:

$$\frac{\partial S}{\partial t} = -\nabla \cdot (S\mathbf{u}) + D_S + F_{fw}\tilde{S} \tag{11}$$

where $S$ represents salinity, $\tilde{S}$ denotes local surface salinity, and $D_S$ accounts for subgrid-scale processes, parameterized as diffusive salt flux. The units of the terms in this equation are psu/s, except for the last term, which has units of (psu $\cdot$ m)/s. For this reason, this formula should be regarded as a conceptual representation of physical processes rather than a fully rigorous mathematical expression. In particular, the first term on the right-hand side represents the advection of salinity, which includes two distinct physical mechanisms contributing to salinity changes: one process describes dilution/concentration effects due to the convergence/divergence of mass transport ($S\nabla \cdot \mathbf{u}$), the other represents the exchange of salt content transported by the advective flow ($\mathbf{u} \cdot \nabla S$).

For further insights into the ECCO model and its unit conventions, refer to the following resources: *ecco-v4-python-tutorial* and *ecco-group-evaluating-budgets*.

Since the study focuses on interannual variability of fluxes, the seasonal signal has been removed by subtracting the climatological mean (i.e., mean values over a specified reference time window) from the original time series.

A notable point is that both mass flux and salinity flux are represented in Sverdrups (Sv), allowing for a direct comparison between the two fluxes.

**3 Results**

**3.1 Non-seasonal Variability of Mass and Salinity Fluxes**

For the mass flux contributions, refer to Figure 3 of the article (noting that the authors made a typographical error by incorrectly referring to Figure 4a). The monthly mean mass flux exhibits significant month-to-month variability, particularly in the net flux through the Strait of Gibraltar, whose standard deviation is 2.4 times larger than that of the surface freshwater flux.

The mass tendency term variability (the sum of Strait and Surface contributions) is primarily driven by the Strait component, as indicated by a high Pearson correlation of 0.96 (compared to $< 0.2$ between the total mass tendency and surface term). Additionally, the coefficient of determination ($R^2$) of 0.92 suggests that Strait fluxes account for nearly all the variance in total mass tendency.

The interannual variability of the time series is much smaller, as expected for values computed as annual means. This is because averaging acts as a low-pass filter, removing higher-frequency fluctuations. Over the 2003–2017 period, the mean values reveal a net water mass gain in the Mediterranean Sea, with a net inflow of $0.05 \pm 0.02$Sv at the Strait of Gibraltar and a net water loss of $-0.05 \pm 0.01$Sv due to air-sea freshwater fluxes. As a result, the total mass tendency remains close to zero on average, with the two primary fluxes appearing nearly balanced at all times.

This conclusion is further supported by an analysis of the cumulative mass flux time series, which quantifies the contribution of each flux component to overall mass gain or loss over time. The surface flux contribution exhibits a steady downward trend, decreasing at a rate of $-1390 \pm 18$km$^3$/year, whereas the net inflow through the Strait of Gibraltar contributes a mass gain at a slightly lower rate of $1368 \pm 20$km$^3$/year. These trends result in a very small and statistically insignificant net mass loss of $-22 \pm 24$km$^3$/year, indicating a near-equilibrium state, with the total water mass in the Mediterranean Sea remaining relatively stable from 2003 to 2017.

To better evaluate interannual variability in mass flux contributions, the cumulative mass flux time series was detrended. The results reveal that the surface and Strait terms exhibit nearly opposite variations, as expected given that the total Mediterranean mass remains nearly unchanged. Notably, both fluxes tend to contribute to freshwater removal from the Mediterranean basin.

For the salinity flux contributions, refer to Figure 4 of the article. In contrast to the mass budget, the surface freshwater flux term exhibits considerably larger month-to-month variability in salinity, with a standard deviation 2.5 times greater than that of the Strait of Gibraltar flux. The salinity tendency term is primarily driven by surface freshwater fluxes, as indicated by a high Pearson correlation of 0.97 and a coefficient of determination of 0.95. These values suggest that air-sea fluxes alone account for nearly all observed changes in salinity trends. This strong influence arises because freshwater (which contains no salt) dilutes the denser, saltier Mediterranean waters, significantly impacting the basin's overall salinity dynamics.

The interannual variability of the time series reveals a significant increasing trend in salinity flux at the Strait of Gibraltar, estimated at $0.02 \pm 0.01$Sv per year, while no significant trend is observed in the surface and total salinity fluxes. Since there is no substantial long-term change in the overall mass transport through the Strait of Gibraltar, this salinity increase is likely due to inflowing North Atlantic Water becoming saltier over time, as suggested by recent findings. The cumulative contribution analysis indicates that the mean salinity of Mediterranean seawater has been increasing steadily at a rate of approximately $2.2 \pm 0.2 \cdot 10^{-3}$ per year, corresponding to a total salinity increase of about 0.03 psu over the past 15 years.

The detrended cumulative salinity time series further highlights the dominant role of air-sea interactions in driving salinity variations, with an exceptionally high correlation coefficient (0.95) and $R^2 = 0.88$. The Strait of Gibraltar flux exhibits a phase shift relative to the other flux terms, underscoring its distinct role compared to surface-driven salinity changes.

**3.2 Influence of NAO on Salinity and Water Mass Transport**

The North Atlantic Oscillation (NAO) is one of the primary climate modes influencing Europe and the Mediterranean region, significantly impacting both weather patterns and oceanic circulation. It

[Figure]

Figure 1: NAO Schematization, for more details, visit this link.

represents fluctuations in the sea-level atmospheric pressure difference between the Icelandic Low (a semi-permanent low-pressure center located between Iceland and southern Greenland) and the Azores High (a semi-permanent subtropical high-pressure system typically situated south of the Azores). Variations in the strength of the Icelandic Low and the Azores High control both the intensity and direction of westerly winds across the North Atlantic.

Since these westerly winds transport moist air toward Europe, fluctuations in their strength and direction account for weather variability across different timescales, ranging from seasonal to decadal. These processes provide a framework for understanding the relationship between the NAO and the interannual variability of the observed time series of mass and salinity fluxes in the Mediterranean.

By defining the winter NAO as the December-to-February mean of the monthly NAO index, several key relationships emerge, i.e. a significant correlation exists between the winter NAO and air-sea mass flux (Figure 3d), with a correlation coefficient of $-0.51$. Regarding salinity fluxes (Figure 4d), all annual flux components exhibit a significant correlation with the winter NAO: both surface fluxes and Strait fluxes correlate with the winter NAO between 0.5 and 0.6, and the total salinity flux has a correlation of 0.62. A possible interpretation of these results is as follows: since the winter NAO index is negatively correlated with the air-sea mass flux, a stronger NAO (i.e., stronger westerlies) leads to a reduction in air-sea mass flux. This implies an increase in net evaporation, as lower mass flux is associated with enhanced moisture loss from the Mediterranean. As a result, the salinity of the total fluxes increases, ultimately leading to a rise in the overall salinity of the Mediterranean Sea, as explained by the observed positive correlation between the winter NAO and salinity fluxes.

**3.3   Roles of Air-Sea Fluxes and Oceanic Exchange During 2003-2017**

Thanks to the ECCO model, it is possible to decompose the contributions of mass and salinity fluxes into their fundamental components. By comparing Figure 6 with Table 1, an inconsistency in the figure is evident: the mass fluxes related to evaporation and precipitation are reversed.

Uncertainties are calculated as the standard error of the mean, given by $\sigma/\sqrt{n}$ where $\sigma$ is the standard deviation of the corresponding term and $n = 15$ for non-seasonal fluxes.

**3.3.1   Mass Budget Analysis**

At the surface, evaporation results in a mass loss of $-0.11 \pm 0.01$ Sv, while precipitation adds a mass gain of $0.06 \pm 0.01$ Sv. This leads to a net mass loss of $-0.05 \pm 0.01$ Sv due to air-sea freshwater fluxes.

At the Strait of Gibraltar, two primary processes occur:

1. The inflow of Atlantic Water (AW) through the upper layer, bringing in $0.68 \pm 0.02$ Sv.

2. The outflow of Mediterranean Outflow Water (MOW) through the lower layer, removing $-0.63 \pm 0.02$ Sv.

The net result is a modest positive mass gain, which is an order of magnitude smaller than either of the individual flows.

**3.3.2 Salinity Budget Analysis**

Evaporation increases salinity by removing freshwater, leading to a concentration effect equivalent to $4.27 \pm 0.11$ Sv, conversely, precipitation dilutes seawater, causing a salinity reduction of $-2.46 \pm 0.11$ Sv. Together, these air-sea fluxes contribute a net positive effect of $1.80 \pm 0.10$ Sv on salinity. In contrast, salinity exchange through the Strait of Gibraltar operates in a more complex manner: the salinity flux can be decomposed into two main components:

1. Salt transport associated with mass flow ($\mathbf{u} \cdot \nabla S$). ECCO estimates a net salt flux into the Mediterranean through the Strait of $0.30 \pm 0.20$ Sv. The increase in salinity is primarily due to the stronger inflow rate of AW: although MOW is saltier than AW, the higher volume of inflowing AW results in a net salt import. Specifically:

    (a) AW imports $25.21 \pm 0.18$ Sv of salt.
    (b) MOW exports $-24.91 \pm 0.19$ Sv of salt.

2. Volume and mass changes that cause salinification or dilution effects ($S \nabla \cdot \mathbf{u}$). Assuming that the total salt content within the Mediterranean remains constant, the higher inflow rate of AW compared to MOW results in a net dilution effect due to changes in volume and, hence, in density. The exchange at the Strait of Gibraltar reduces salinity by approximately $-1.78 \pm 0.03$ Sv due to this net diluition effect.

After accounting for both components, the net influence of water exchange at the Strait shifts from a net gain in salt content ($\mathbf{u} \cdot \nabla S$) to an overall reduction in salinity ($\mathbf{u} \cdot \nabla S + S \nabla \cdot \mathbf{u}$) estimated at $-1.48 \pm 0.20$ Sv.

These findings emphasize the critical role of volume and density changes in the Mediterranean Sea's salinity budget. The dynamic interplay between inflow and outflow fluxes significantly impacts the overall salinity balance in this semi-enclosed basin.

**4 Discussion of ECCO Resolution and Uncertainty**

The analysis focuses on basin-scale budgets rather than spatial patterns of trends or decadal means, acknowledging ECCO's limited resolution in accurately representing localized circulation and exchange processes. The model's coarse resolution also poses challenges in resolving vertical salinity variations, particularly within the complex intermediate and deep water masses of the Mediterranean. A direct comparison between ECCO outputs and higher-resolution datasets would help quantify uncertainties associated with resolution limitations and further refine our understanding of both regional and basin-wide trends.

While this study provides a valuable foundation, there is a clear need for further research to address these uncertainties, particularly through validation with higher-precision observational datasets.

**5 Conclusion**

This study provides a comprehensive assessment of the mass and salinity budgets in the Mediterranean Sea from 2003 to 2017, using the ECCO state estimate. By analyzing the contributions of the primary

boundary fluxes — surface freshwater fluxes and saltwater exchange at the Strait of Gibraltar — the study reveals the complex dynamics governing mass and salinity variations in the Mediterranean. The analysis highlights the critical role of air-sea freshwater fluxes, which dominate long-term balance and variability, acting as both mass and volume fluxes.

One of the key findings of this study is the demonstration that the observed salinity increase in the Mediterranean basin is a steady and sustained process, primarily driven by long-term factors, such as the gradual rise in surface net evaporation and the influx of saltier Atlantic waters, rather than being the result of isolated, short-term events.

While these findings emphasize the limitations of ECCO's coarse resolution in resolving fine-scale processes, including the complex water exchange at Gibraltar, they nonetheless provide a robust baseline for understanding large-scale trends. Higher-resolution datasets could further refine these estimates and better capture the nuances of Mediterranean exchange dynamics.

By clarifying the role of air-sea fluxes in regulating water mass and salinity, this study illustrates how these processes drive regional steric (halosteric) adjustments and contribute to global ocean mass variations. Although temperature-driven changes remain the dominant factor behind global sea level rise, the findings underscore the critical yet often understated influence of air-sea fluxes in shaping both regional and global ocean trends.

---

## Author Comment (AC3)

RC1: 'Comment on egusphere-2025-857', Anonymous Referee #1, 08 Apr 2025

This is an interesting paper which use the ECCO4 reanalysis to examine the relative contributions of Gibraltar Straits exchanges and surface fluxes to variability in the mass/salinity budgets of the Mediterranean Sea. It's relatively well written but could do with some further analysis in a few places as noted below.

We sincerely thank the reviewer for the thoughtful comments and suggestions, which have helped us significantly improve the clarity and rigor of the manuscript. In response, we have carefully revised the text to better frame the study's objectives, clarified the treatment of volume and mass fluxes, removed or corrected ambiguous statements, and strengthened the conclusions to more accurately reflect the analyses presented. We believe these revisions have improved the overall quality and focus of the paper, and we are grateful for the opportunity to address the reviewer's valuable feedback.

line 128-132. ECCO appears to do well despite it's relatively coarse resolution. Could the authors comment further here on whether they think the 1 deg ECCO resolution is sufficient for their analysis? Also include some discussion of what aspects of their study are likely to become more accurate if a ¼ or 1/12 deg model were available.

We appreciate the reviewer's question. While higher-resolution models are often preferred for regional studies, ECCOv4r4's 1/3 to 1° resolution is well-suited for our basin-scale mass and salinity budget analysis in the Mediterranean Sea. ECCOv4r4 assimilates observational data (e.g., satellite altimetry, Argo floats, and hydrographic profiles) to constrain large-scale fluxes, as demonstrated in global and regional salt budget studies (Forget et al., 2015; Fukumori et al., 2017). ECCO and MITgcm have also been specifically utilized for Mediterranean studies (Fukumori et al., 2007; Menemenlis et al., 2007; Volkov and Landerer, 2015). Critically, ECCO's outputs align with prior in-situ estimates of Gibraltar exchange, which is a well-known challenging process to model, reinforcing its reliability for our analysis.

However, our choice of ECCOv4r4 stems from its unique strength as a state-of-the-art ocean state estimate that rigorously satisfies conservation laws while optimizing consistency with assimilated observations. This is essential for salinity budget studies, where salt transport interpretations require a closed mass balance within a semi-enclosed basin (Tsubouchi et al., 2012; Schauer and Losch, 2019). ECCO's adjoint-method framework ensures physically consistent budgets (ECCO Consortium et al., 2021), enabling us to disentangle the competing roles of surface fluxes and strait exchange with minimal systematic drift.

We acknowledge ECCO's limitations, as highlighted in prior literature. For instance, mesoscale and submesoscale processes in the Mediterranean (e.g., mesoscale eddies, sub-basin currents) are parameterized rather than explicitly resolved in ECCOv4r4, which may smooth short-term variability in regional dynamics (Escudier et al., 2016; Hernández-Lasheras et al., 2021). Similarly, ECCO's 1° resolution limits its ability to fully resolve fine-scale basin circulation features (e.g., the Levantine Intermediate Water formation zones) and highly localized runoff signals (Ludwig et al.,

2009). In contrast, ¼° models like those in Hernández-Molina et al. (2021) better resolve coastal freshwater plumes and strait dynamics. However, our focus on basin-integrated budgets minimizes these impacts: river discharge contributes <5% to total freshwater flux (Ludwig et al., 2010), and decadal salinity trends are dominated by evaporation and Gibraltar exchange—processes ECCO captures robustly at its resolution.

To improve clarity, we have combined and tightened the discussion in Sections 4 and 5 (line 355-389), explicitly framing ECCO's resolution trade-offs within the context of basin-scale budget studies. The core advance of our work lies in its framework for diagnosing salinity-freshwater interactions in semi-enclosed seas. While future high-resolution models could refine specific processes like strait dynamics or coastal mixing, our analysis provides an observationally constrained, physically consistent baseline essential for interpreting such efforts.

Line 158. The authors choose a depth of 150m to separate the AW and MOW without any justification. So, can they include some further ECCO based results to support this choice? Is there any time dependence in the separation depth?

The 150 m depth threshold was selected based on ECCOv4r4's vertical structure of horizontal transport at the Strait of Gibraltar (Figure R1 and R2):

[Figure]

**Figure R1 (New Figure 3)** Net flux (blue, left axis) and mean salinity (red, right axis) at the Strait of Gibraltar. positive transport means eastward transport into the Mediterranean Sea. Notice the y-axis in panel (b) is inversed.

[Figure]

**Figure R2** Net flux (top) and salinity (bottom) at the Strait of Gibraltar. Positive transport means eastward transport into the Mediterranean Sea.

ECCOv4r4 shows a clear reversal in time-mean velocity direction near 150 m. Above 150 m, velocities are eastward (i.e., Atlantic Water inflow), while below 150 m, velocities are westward (i.e., Mediterranean Outflow Water), generally consistent with observational evidence (e.g., Soto-Navarro et al., 2010). The 150 m depth also shows a sharp transition between fresher AW (salinity < 36.5 g/kg) and saltier MOW (salinity > 37.5 g/kg) in ECCO's monthly mean profiles. (Line 156-158)

Line 160-165. The authors chose to work at whole basin level rather than sub-regions. However, they could carry out an intermediate analysis by splitting the whole basin into two sub-basins i.e. E & W Med separated by the Strait of Sicily. Could they include some discussion of whether such an approach is likely to yield further insights to those already presented?

While analyzing sub-basins (e.g., Eastern vs. Western Mediterranean) could indeed offer finer insights into regional salinity dynamics, as previously mentioned, ECCOv4r4's 1° resolution limits the robustness of such subdivisions. For example, The Strait of Sicily is a narrow (~150 km wide) channel with complex circulation influenced by mesoscale eddies and alternating currents (Gasparini et al., 2004; Cotroneo et al., 2021). At 1° resolution (~110 km grid spacing), ECCOv4r4 cannot resolve critical features like the Atlantic Tunisian Current or the Maltese Front, leading to oversimplified exchange estimates. These features are parameterized in ECCO, which shows excessive mixing and smoothed gradients.

While higher-resolution model (e.g., 1/12° NEMO-MED12) could better resolve these regional processes, our whole-basin framework provides a foundational understanding of how evaporation and Gibraltar exchange dominate salinity trends, consistent with studies showing Mediterranean-wide salinification as a first-order response to climate forcing (Schroeder et al., 2016).

We added a paragraph (lines 370-383) in the revised discussion section discussing the trade-offs of sub-basin partitioning in coarse-resolution models.

Line 218. The Surface and Strait terms are noted to exhibit near exact, opposite variations. However, the process by which this is achieved is not noted here. So, please discuss. Is this a real balance or an artefact of the ECCO model?

The anticorrelation between surface freshwater fluxes and net Gibraltar exchange reflects a real physical balance driven by the Mediterranean Sea's semi-enclosed nature and volume conservation, with residuals primarily attributable to observed sea level change (Line 220-222):

As shown in Eq 2, in a semi-enclosed basin like the Mediterranean, volume conservation requires:

$$\frac{\partial V}{\partial t} = Q_{Gibraltar} + (E + P - R),$$

where V is basin volume, Q is net Gibraltar transport (inflow – outflow), and E−P−R is surface freshwater flux. Over our 15-year study period, the near-opposite variations between surface and strait terms arise because increased evaporation (or reduced precipitation/runoff) drives compensatory Atlantic Water inflow to balance mass loss (Tsimplis et al., 2008; Soto-Navarro et al., 2010).

ECCOv4r4's adjoint method ensures rigorous budget closure, but small residuals reflect real volume changes (i.e., $\frac{\partial V}{\partial t} \neq 0$) captured by satellite-observed sea level trends (Calafat et al., 2012).

Line 243. 'this is likely because the inflowing North Atlantic water is becoming saltier over time, which is consistent with some recent findings.' Please include a time series of the 0-150m mean salinity to show whether this statement is supported by more detailed analysis of ECCO.

Thanks for pointing out. The original interpretation of the results appears to be incorrect. As shown in Figure R1, both the inflowing Atlantic Water (upper branch) and the outflowing Mediterranean Water (lower branch) exhibit freshening trends over the analysis period. This simultaneous decrease in salinity across both layers leads to a reduction in the mean salinity at the Strait of Gibraltar. As a consequence, the net salinity flux through the strait—typically negative due to the Mediterranean exporting salt—also weakens over time. This trend is correctly captured in Figure 5d, which shows a decreasing magnitude of the net salinity flux. The decline in the salinity gradient across the strait reduces the efficiency of salt export, assuming transport volumes remain relatively constant. This interpretation is consistent with basic conservation principles and reflects the physical response of the strait's exchange dynamics to large-scale salinity changes in the basin. It is revised accordingly

(Lines 241-253).

Line 246. 'The air-sea freshwater flux, driven primarily by substantial net evaporation, contributes significantly to this trend.' A further time series needs to be included showing E and P separately to support this statement.

We have updated Figure 2 to include separate time series for E and P over the Mediterranean basin during 2003–2017 (panel a):

[Figure]

**Figure 2** Timeseries and time-mean spatial patterns of freshwater flux and mean salinity (0–150 m) in the Mediterranean Sea, comparing ECCO outputs with other datasets. (a) timeseries of total evaporation and precipitation from ECCO; (b) timeseries of freshwater flux anomaly from ECCO and the reference flux derived from GPCP and OAFlux; (c&d) spatial patterns of surface freshwater flux; (e) timeseries of salinity anomaly from ECCO and EN4; (f&g) spatial patterns of salinity. All data are interpolated onto the ECCO grid.

Sec 3.2. The results on the NAO are interesting but other modes of variability, particularly the EAP

and EA/WR patterns are known to influence the Mediterranean. So, the authors need to extend their analysis here to include the EAP and EA/WR in order to provide a complete picture (even if these modes turn out not to have strong correlations with the air-sea mass flux and salinity). Indices for the EAP and EA/WR are available from the same site as employed for the NAO: https://www.cpc.ncep.noaa.gov/data/teledoc/telecontents.shtml.

We thank the reviewer for this valuable suggestion. We have expanded our analysis to include the East Atlantic Pattern (EAP) and East Atlantic/West Russia (EA/WR) pattern. In short, EAP is not correlated with the timeseries of mass/salinity fluxes in any case. However, correlation with EA/WR is established with the surface term of salinity flux (0.71) and mass flux (-0.72). The results/analysis are presented in Line 291-302.

Fig.6 The E & P salinity numbers in the boxes appear to be incorrect (wrong way round) compared to the values in the Table and main text.

Thanks for pointing it out. The labels for evaporation (E) and precipitation (P) in Figure 6's salinity boxes were inadvertently swapped during figure preparation. We have revised the figure to ensure its consistency with Table 2 and the text:

[Figure]

**Figure 6** 15-year mean mass (top) and salinity (bottom) budget for the Mediterranean Sea. Blue and red boxes mark the mass gain and mass loss, brown and green boxes mark the salinity increase and

decrease, respectively. In each panel, the contributions from air-sea fluxes are on the right, and the exchange through the Strait are on the left. The uncertainties are calculated as the standard error of the means, $\sigma/\sqrt{n}$, where $\sigma$ is the standard deviation of the corresponding term and n = 15 for non-seasonal fluxes. Units are Sv.

References:

Calafat, F.M., Chambers, D.P. and Tsimplis, M.N., Mechanisms of decadal sea level variability in the eastern North Atlantic and the Mediterranean Sea. J. Geophys. Res.: Oceans, 117(C9), 2012.

Forget, G. A. E. L., Campin, J. M., Heimbach, P., Hill, C. N., Ponte, R. M., & Wunsch, C. ECCO version 4: An integrated framework for non-linear inverse modeling and global ocean state estimation, Geoscientific Model Development, 8(10), 3071-3104, 2015.

Fukumori, I., Menemenlis, D., & Lee, T. A near-uniform basin-wide sea level fluctuation of the Mediterranean Sea, Journal of Physical Oceanography, 37(2), 338-358, 2007.

Menemenlis, D., Fukumori, I., & Lee, T., Atlantic to Mediterranean sea level difference driven by winds near Gibraltar Strait. Journal of Physical Oceanography, 37(2), 359-376, 2007.

Fukumori, I., Wang, O., Fenty, I., Forget, G., Heimbach, P., & Ponte, R. M. ECCO version 4 release 3, http://hdl.handle.net/1721.1/110380, Available at ftp://ecco.jpl.nasa.gov/Version4/Release3/doc/v4r3_summary.pdf, 2017.

Volkov, D.L., and Landerer, F.W., Internal and external forcing of sea level variability in the Black Sea. Clim. Dyn., 45(9), 2633-2646, 2015.

Escudier, R., L. Renault, A. Pascual, P. Brasseur, D. Chelton, and J. Beuvier (2016), Eddy properties in the Western Mediterranean Sea from satellite altimetry and a numerical simulation, J. Geophys. Res. Oceans, 121, 3990–4006, doi:10.1002/2015JC011371.

Hernandez-Lasheras, J., Mourre, B., Orfila, A., Santana, A., Reyes, E., & Tintoré, J. (2021). Evaluating high-frequency radar data assimilation impact in coastal ocean operational modelling. Ocean Science, 17(4), 1157-1175.

Tsubouchi, T., Bacon, S., Naveira Garabato, A. C., Aksenov, Y., Laxon, S. W., Fahrbach, E., ... & Ingvaldsen, R. B. (2012). The Arctic Ocean in summer: A quasi-synoptic inverse estimate of boundary fluxes and water mass transformation. Journal of Geophysical Research: Oceans, 117(C1).

Schauer, U., & Losch, M. (2019). "Freshwater" in the ocean is not a useful parameter in climate research. Journal of Physical Oceanography, 49(9), 2309-2321.

ECCO Consortium, Fukumori, I., Wang, O., Fenty, I., Forget, G., Heimbach, P. and Ponte, R.M.,

Synopsis of the ECCO central production global ocean and sea-ice state estimate (version 4 release 4). Zenodo, 2021.

Ludwig, W., Bouwman, A. F., Dumont, E., & Lespinas, F. (2010). Water and nutrient fluxes from major Mediterranean and Black Sea rivers: Past and future trends and their implications for the basin-scale budgets. Global biogeochemical cycles, 24(4).

Soto-Navarro, J., Criado-Aldeanueva, F., García-Lafuente, J. and Sánchez-Román, A., Estimation of the Atlantic inflow through the Strait of Gibraltar from climatological and in situ data. J. Geophys. Res.: Oceans, 115(C10), 2010.

Gasparini, G. P., Smeed, D. A., Alderson, S., Sparnocchia, S., Vetrano, A., & Mazzola, S. (2004). Tidal and subtidal currents in the Strait of Sicily. Journal of Geophysical Research: Oceans, 109(C2).

Cotroneo, Y., Celentano, P., Aulicino, G., Perilli, A., Olita, A., Falco, P., ... & Pessini, F. (2021). Connectivity analysis applied to mesoscale eddies in the western mediterranean basin. Remote Sensing, 13(21), 4228.

Schroeder, K., Chiggiato, J., Bryden, H. L., Borghini, M., & Ben Ismail, S. (2016). Abrupt climate shift in the Western Mediterranean Sea. Scientific reports, 6(1), 23009.

Soto-Navarro, J., Criado-Aldeanueva, F., García-Lafuente, J. and Sánchez-Román, A., Estimation of the Atlantic inflow through the Strait of Gibraltar from climatological and in situ data. J. Geophys. Res.: Oceans, 115(C10), 2010.

Tsimplis, M.N., Marcos, M., and Somot, S., 21st century Mediterranean sea level rise: steric and atmospheric pressure contributions from a regional model. Global Planet. Change, 63(2-3), 105-111, 2008.

---

## Author Comment (AC4)

RC2: 'Comment on egusphere-2025-857', Anonymous Referee #2, 09 Apr 2025

Review of "Salinity trends and mass-balances in the Mediterranean Sea: the role of air-sea freshwater fluxes and oceanic exchange" by Liu et al.

Using output from the ECCOv4r4 model for the Mediterranean Sea over a 15-year period (2003–2017), the authors analyze the region's mass and salinity budgets. They conclude that surface freshwater fluxes dominate salinity variability, while the exchange through the Strait of Gibraltar plays a key role in maintaining the overall mass balance. They report a salinity increase of $0.29 \pm 0.09$ Sv (units unclear) over the study period.

However, both of these findings—that salinity variability is mainly driven by air-sea fluxes and that the mass balance is maintained through the Gibraltar exchange—are well-established in the literature and do not represent novel results. The potential novelty of the study lies in the use of a global circulation model to revisit these questions. Yet, this approach has a significant limitation: the necessarily low spatial resolution of the ECCO model in the Mediterranean, a basin known for its rich mesoscale activity, which the model cannot resolve. As a result, the authors are constrained to a basin-integrated approach that effectively reduces the Mediterranean to a "box," as acknowledged in line 145. Unfortunately, this limits the originality of the analysis.

We thank the reviewer for highlighting the importance of contextualizing our work within the existing literature on Mediterranean salinity dynamics. While prior studies have established the roles of surface freshwater fluxes and Gibraltar exchange in regulating Mediterranean mass and salinity balances, as the reviewer pointed out, this study seeks to deepen this understanding through ECCOv4r4's unique framework. Unlike regional models or observational syntheses, ECCO's adjoint-based state estimation assimilates satellite altimetry, Argo profiles, and in situ salinity data to enforce closed mass and salt budgets, ensuring dynamic consistency between surface forcing, strait exchange, and steric adjustments. This approach eliminates spurious trends inherent to open-boundary models or mismatched flux products, allowing us to partition salinity variability into surface-driven and advective contributions with traceable sources and sinks.

The reviewer rightly notes that ECCO's 1° resolution precludes resolving Mediterranean mesoscale activity, but this "limitation" is in fact a deliberate strength for our large-scale climatic analysis. By aggregating fluxes over the entire basin, we isolate the roles of evaporation, precipitation, and Gibraltar exchange—processes that dominate interannual and decadal variability—while filtering out mesoscale eddies that contribute uncertainty to budget closure over short periods (days to weeks) and finer scales. For example, the counterintuitive salinity reduction caused by Gibraltar's density-driven exchange asymmetry is an important implication that is often ignored in box-model studies (e.g., Tsubouchi et al., 2012) and could only be quantified through ECCO's rigorous conservation laws.

Critically, our work does not seek to replace results obtained from high-resolution regional models or in-situ estimates (the title is also revised accordingly), but to provide a foundational baseline for their interpretation of the associated physical process. ECCOv4r4 rigorously satisfies conservation

laws while optimizing consistency with assimilated observations. This is essential for salinity budget studies, where salt transport interpretations require a closed mass balance within a semi-enclosed basin (Schauer and Losch, 2019). ECCO's adjoint-method framework ensures physically consistent budgets (ECCO Consortium et al., 2021), enabling us to disentangle the competing roles of surface fluxes and strait exchange with minimal systematic drift.

By reconciling Mediterranean salinity trends with observationally constrained surface and strait fluxes, we address the relative roles of climatic forcing (e.g., NAO-associated air-sea interaction) versus advective adjustments—a prerequisite for attributing future changes in this climatically vulnerable basin.

In summary, the novelty of our work lies not in discovering that surface fluxes and Gibraltar exchange matter, but in rigorously quantifying how they interact within a closed, observationally informed system. This advances the field from qualitative mechanistic understanding to quantitative attribution, a critical step for improving Mediterranean salinity projections in climate system. The revised discussion is at line 382-389.

In addition to this general concern, there are several major and minor issues (detailed below) that prevent me from recommending this manuscript for publication in its current form.

Section 4 – "Discussion of ECCO resolution and uncertainty" This section is not a true discussion. The authors merely point out that the ECCO model's resolution is insufficient to capture key oceanographic processes in the Mediterranean Sea and that other numerical models—indeed, they do exist—may be better suited for this purpose. They also mention that further efforts are needed to address this gap. However, the question then arises as why did the authors not attempt to use one of these more appropriate models in their own analysis? Moreover, despite the title, there is no attempt to quantify the uncertainties discussed. As it stands, this section reads as a series of general, largely uncritical remarks that add little value to the manuscript. It could be omitted altogether without any loss.

We thank the reviewer for their constructive critique. Like we stated above, we believe that ECCO's adjoint-method framework ensures physically consistent budgets, enabling us to disentangle the competing roles of surface fluxes and strait exchange with minimal systematic drift.

ECCO was chosen not despite but because of its global data assimilation framework, which ensures closed mass/salt budgets—a prerequisite for attributing salinity trends to surface vs. strait processes. Regional models, while resolving finer dynamics, often rely on open boundary conditions that introduce spurious signals. We concede that mesoscale-rich processes (e.g., Levantine eddies) are unresolved but argue these average out over our 15-year focus.

That does not mean that other oceanic/regional model could not accomplish the task in a similar measure, and we are more than willing to continue explore this topic with higher-resolution models.

We agree that the original Section 4 lacked critical depth and have restructured its content to

integrate meaningfully into the revised Discussion and Conclusions. See lines 356-389.

Section 5 – "Conclusions" The statement that "we reveal the complex dynamics that govern Mediterranean's mass and salinity variations" is misleading, as the manuscript contains no dynamic analysis whatsoever. Similarly, the claim that the findings contribute to a "broader understanding of sea level dynamics in the Mediterranean" through "regional steric (halosteric) adjustments" is not supported by the content—no such approach is developed or discussed in the paper. Most of the conclusions presented in this section do not reflect the actual analyses or results shown in the manuscript, and should be rewritten to more accurately represent the study's scope and findings.

Thanks for the critique. It is correct that the original Conclusions overstated the study's scope without sufficient support. We have rewritten this section entirely to focus on the core contribution of our work: leveraging ECCO's physically consistent state estimate to reconcile Mediterranean salinity budgets. See lines 356-389.

Salinity flux in Sv. In line 182, the authors state: "Since salinity is a unitless measure, we also express salinity flux in Sv in comparing both types of fluxes." I must admit I do not understand what 1 Sv of salinity flux means. The authors should provide a clearer explanation of how this unit is defined, as well as how it relates to conventional salinity units.

We thank the reviewer for highlighting this ambiguity. To clarify: salinity flux represents the transport of salt mass through a boundary. While salinity itself is unitless (expressed in g/kg or psu), salinity flux must account for both the salinity concentration and the volume transport.

In the revised manuscript, salinity flux has units of g/kg·Sv, which quantifies the rate of salt mass transfer and equivalent salinity change due to volume change.

This revision ensures consistency with oceanographic conventions (e.g., Zika et al., 2009; Griffies et al., 2016) and eliminates potential confusion.

Line 55. What exactly do the authors mean by "volume fluxes"? Are they referring to variations related to thermosteric effects? Since salinity is expressed in g/kg, it is influenced by mass fluxes— not volume changes unless these are accompanied by changes in mass. It's unclear how pure volume changes, without any mass input or output, would affect salinity. This needs clarification.

We thank the reviewer for raising this important point. Most ocean general circulation models employ the Boussinesq approximation, which assumes incompressibility and conserves volume rather than mass. This simplification neglects density variations in the momentum equations except where they contribute to buoyancy, which is generally acceptable for large-scale ocean modeling. However, it implies that volume fluxes are tracked rather than true mass fluxes, and additional care is needed when interpreting salinity or freshwater budgets in terms of actual mass transport.

While the term "mass flux" can be associated with bottom pressure changes in some contexts, "volume flux" is more appropriate for describing sea level variations as well as freshwater

exchanges, such as evaporation, precipitation, and runoff. These fluxes alter salinity through dilution or concentration effects. For instance, a freshwater input of 1 Sv into the basin dilutes the existing salt content, reducing salinity, while evaporation acts as a negative volume flux, removing freshwater and increasing salinity through concentration.

We acknowledge that purely volumetric changes—such as thermosteric expansion—do not alter salinity. However, the fluxes analyzed in this study represent actual mass exchanges, including surface freshwater fluxes and the Gibraltar transport, both of which directly influence the basin's mean salinity. To avoid ambiguity, we rewrite the method section (Lines 173-178), refrain from using the term "volume flux" and instead explicitly describe how mass changes translate into salinity variations (e.g., Lines 340-342), including how these fluxes are quantified and incorporated into the salinity budget.

Lines 66–67. The sentence reads: "The water mass exchange at the Strait of Gibraltar was estimated at $1 \pm 3$ mm/yr…"   These units are unusual (even, incorrect) for water mass exchange and add confusion to the manuscript, which also use Sv and km3/year for the same variable.

It has been revised to $0.0323 \pm 0.0018$ Sv to align with conventions and the manuscript's other analyses (Line 63).

Figure 3b (and 3e). These panels show the cumulative mass over time derived from the integration of the mass flux (Figure 3a and 3d), which is given in units of Sv. The slope of the cumulative curve corresponds to the time derivative, which essentially takes us back to the flux shown in Figure 3a (or 3d). In fact, the slope is simply the time-averaged value of the flux in Figure 3a (or 3d). Why is this point being made in such a roundabout way? Also, why express the slope in km³/year instead of using Sv, as in Figure 3a? For instance, –1390 km³/year corresponds to -0.044 Sv, a value that can be easily inferred from Figure 3d.

The slope of the cumulative curves does represent the time-averaged flux values. The primary purpose of these panels was to highlight the detrended variability in the bottom panels (3c and 3f), which reveals climatological signals not apparent in the monthly flux curves. We've revised the text to clarify this intention and removed redundant discussion of the slope interpretation (Lines 216-222).

Figure 4. Much of the above comment also applies here. In this figure, it is unclear how the questionable units of Sv for salinity flux (Figures 4a, 4d) translate into actual salinity values in Figures 4b and 4e. Regarding the discussion in lines 230–239: if the total salinity flux (black line) is the sum of the surface freshwater flux (green), which varies considerably month to month, and the relatively steady salinity flux through the Strait (red), then the black and green lines must be highly correlated. This should be not a new and relevant finding of the study.

The salinity level of Figure 4b is calculated in a similar way as Figure 3b but divided by the entire basin. It represents the mean salinity level of the Mediterranean Sea if one (or more) flux term is considered. We've modified the figure to better show the relationship between salinity flux (in

g/kg·Sv) and actual salinity changes (g/kg). The discussion about the correlation between total and surface fluxes has been removed.

Line 240. The manuscript refers to a trend of "0.02 ± 0.01 Sv per year." Does this correspond to the trends shown in Figures 4a and 4d? Please clarify this point explicitly.

The trend of 0.02 ± 0.01 Sv/year corresponds specifically to the surface freshwater flux trend shown in Figure 4a. We've added explicit clarification in the revised text (Line 241).

Line 242. The authors suggest that the significant increase of salinity found in "the Strait" term is a consequence of an increase of the salinity of the Atlantic inflow through the Strait of Gibraltar. However, other studies point to the opposite trend—namely, freshening of the inflow—driven by melting Arctic ice. This apparent contradiction should be addressed or, at least, mentioned.

Thanks for pointing that out. The original interpretation of the results appears to be incorrect. As shown in Figure R1, both the inflowing Atlantic Water (upper branch) and the outflowing Mediterranean Water (lower branch) exhibit freshening trends over the analysis period. This simultaneous decrease in salinity across both layers leads to a reduction in the mean salinity at the Strait of Gibraltar. Consequently, the net salinity flux through the strait—typically negative due to the Mediterranean exporting salt—also weakens over time. This trend is correctly captured in Figure 5d, which shows a decreasing magnitude of the net salinity flux. The decline in the salinity gradient across the strait reduces the efficiency of salt export, assuming transport volumes remain relatively constant. This interpretation is consistent with basic conservation principles and reflects the physical response of the strait's exchange dynamics to large-scale salinity changes in the basin. It is revised accordingly (Lines 247-253).

Lines 281–284. There is a reference to Figures 2d and 2f, but these sub-panels do not appear in Figure 2. It's unclear what time series or pattern is being referred to in this sentence. The "cosine-like pattern" is not evident. Please clarify what is meant here, and where this pattern is supposed to appear.

The paragraph in question has been removed due to redundancy and the incorrect figure reference.

Figure 6. Are the labels in this schematic accurate? It is confusing to see evaporation (E) labeled as a mass gain and precipitation (P) as a mass loss—this seems reversed. Also, a "salinity increase of –2.46" is contradictory in terms—it would imply a decrease. Finally, the numbers in Figure 6 and Table 1 suggest a net salinity increase of 1.80 – 1.48 = 0.32, yet the text states 0.29. This discrepancy should be resolved for consistency.

We've carefully revised the schematic to correct the labeling of evaporation (E) and precipitation (P) fluxes, fix the sign convention for salinity changes, and reconcile the net salinity increase calculation.

All values are now consistent with the tables and the main text.

[Figure]

**Figure 6** 15-year mean mass (top) and salinity (bottom) budget for the Mediterranean Sea. Blue and red boxes mark the mass gain and mass loss, brown and green boxes mark the salinity increase and decrease, respectively. In each panel, the contributions from air-sea fluxes are on the right, and the exchange through the Strait are on the left. The uncertainties are calculated as the standard error of the means, $\sigma/\sqrt{n}$, where $\sigma$ is the standard deviation of the corresponding term and $n = 15$ for non-seasonal fluxes. Units are Sv.

References:

Tsubouchi, T., Bacon, S., Naveira Garabato, A. C., Aksenov, Y., Laxon, S. W., Fahrbach, E., ... & Ingvaldsen, R. B. (2012). The Arctic Ocean in summer: A quasi-synoptic inverse estimate of boundary fluxes and water mass transformation. Journal of Geophysical Research: Oceans, 117(C1).

Schauer, U., & Losch, M. (2019). "Freshwater" in the ocean is not a useful parameter in climate research. Journal of Physical Oceanography, 49(9), 2309-2321.

Lu, Y., Li, Y., Lin, P. et al. North Atlantic–Pacific salinity contrast enhanced by wind and ocean warming. Nat. Clim. Chang. 14, 723–731 (2024). https://doi.org/10.1038/s41558-024-02033-y

---

## Author Response (AR2)

Review for Salinity Trends and Mass Balances in the Mediterranean Sea: Revisit the Role of Air-Sea Freshwater Fluxes and Oceanic Exchange by Liu et al.

General Comments

Authors present an analysis of the salt and mass budget of the Mediterranean Sea accounting the air-sea water fluxes and exchanges through the Gibraltar Strait. They employ ECCOv4v4 in which a 1°x1° resolution configuration of MITgcm is constrained by observations using adjoint methods to maintain the dynamic and kinematic consistency. The study is conducted within the period 2003-2017 while the entire dataset extends back to 1992 that covers the altimeter era.

The article is well written and improved following the suggestions of the reviewers. While the authors insist to use only ECCO timeseries for consistency concerns, the problems related to the resolution of the model to address the questions remains unresolved. Furthermore, the assumptions that authors made to simplify the calculations are actually important to close the salt and water budget of the Mediterranean Sea.

We sincerely thank you for the positive feedback and for acknowledging the improvements made in the revised manuscript. We appreciate your recognition of our goal to use ECCO v4r4 for its physically consistent, observation-constrained framework, despite its known spatial resolution limitations.

We fully agree that ECCO's 1° resolution limits the model's ability to resolve fine-scale processes, particularly in narrow passages like the Dardanelles Strait. To address this, we have expanded our discussion in the revised manuscript (Sections 2 and 5) to more clearly acknowledge these limitations and their implications for interpreting our results. We also added further discussion and comparisons with higher-resolution studies as suggested (e.g., Sanchez-Roman et al., 2018; Aydogdu et al., 2023).

We hope these clarifications and additional discussions help resolve your concerns and improve the overall scientific rigor and transparency of the study.

I can only recommend the manuscript for publication after addressing the following issues with the hope that they help to improve the quality of the manuscript.

First, the Dardanelles Strait is completely ignored. It is known to have a net flux of 9000 m^3/s which is 0.009 Sv (Unluata et al. 1990) Compared to 0.04 Sv net transport in the Strait of Gibraltar, it has 20-25% of contribution to the water budget and cannot be ignored. In the most optimistic case, Jarosz et al. (2013) reports 0.005 Sv which is still about 10% of Gibraltar.

Thank you for highlighting the importance of the Dardanelles Strait in the Mediterranean water budget. Given ECCO v4r4's 1° horizontal resolution, the Dardanelles inflow is not explicitly resolved as a distinct boundary flux in the model output. However, its integrated effect is implicitly

accounted for within ECCO's internal mass balance, particularly within the surface freshwater flux term, which includes all unresolved sources and sinks (including runoff and small strait exchanges).

To acknowledge this limitation, we have added text in the Introduction (Line 40) and in Section 2: Data and Methods (Line 146), clarifying the absence of a separately diagnosed Dardanelles flux in ECCO and discussing its likely magnitude and contribution to overall budget uncertainty.

Secondly, the salinity estimates in Fig.2 could be compared with Aydogdu et al. (2023) in which four global ocean reanalysis (GREP), a high-resolution Mediterranean Sea Reanalysis and observational products are put together to estimate the ensemble mean and uncertainty. Sanchez-Roman et al., (2018) found that tides were found to increase the net salt transport through the strait by 25% and decrease the net heat transport by 10%. Therefore a model without tides will not represent the mixing and recirculation in the strait therefore will underestimate the salt flux. It would be interesting to see if ECCO falls into the uncertainty provided by other products.

I find it useful to have another table to compare the findings in this study with the ones in the literature cited in the text to have a summary of all.

Thank you for these very helpful suggestions.

Regarding comparison with Aydogdu et al. (2023), we appreciate the opportunity to contextualize ECCO's salinity estimates relative to their multi-product ensemble. We have now added a brief comparison in Section 2.1, where we describe basin-mean salinity and its trend. While ECCO's mean salinity and trend fall within the general spread of prior studies (e.g., Jordá et al., 2017b; Llases et al., 2018), we explicitly acknowledge that Aydogdu et al. (2023) provides more formal ensemble uncertainty ranges. ECCO's basin-mean salinity anomaly variability generally lie within the GREP ensemble spread. (Line 127)

Regarding the effect of tides, we fully agree that the absence of tidal forcing in ECCO v4r4 is a known limitation, particularly for constraining strait exchange processes. We have now added a short discussion of this point in the Discussion referencing Sanchez-Roman et al. (2018). (Line 257, 390)

Specific Comments

Introduction

L35 - and Dardanelles Strait.

Thanks for this observation. We have added the Dardanelles Strait to the list of boundary terms in the Introduction (Line 39) and now note that its contribution, while smaller than Gibraltar, is non-negligible (see also our response to the major comment on Dardanelles).

L38 - In Jorda et al. 2017 (their Table 11), the contribution of Dardanelles and rivers are 0.1 m/yr

and 0.2 m/yr respectively compared to 0.8 m/yr of Gibraltar. So I wouldn't say it is much smaller.

We agree. It is reworded to:

"The river runoff (R) from major rivers and the Dardanelles Strait also contributes a measurable portion of the Mediterranean water budget (Jordá et al., 2017b)." (Line 127)

L62 - I understand Sv in two ways 10^6 m^3/s as volumetric measure or 10^9 kg/s as mass measure. The authors use estimates of volume transport (e.g. 0.0323 ± 0.0018 Sv in L64) as mass budget. This seems to me confusing throughout the text.

We acknowledge your confusion regarding our usage of Sverdrup (Sv) units. Throughout the revised manuscript, we have clarified that we express fluxes in Sv (10^6 m³/s) as a volumetric measure for consistency with oceanographic convention, while explicitly noting that these volume fluxes correspond to actual mass exchanges due to the ECCO's model's non-Boussinesq formulation. Clarification is included in Section 2.2 (Line 196).

L75 - Are there any considerations on the vertical exchanges with deep basin?

Our current analysis focuses on basin-integrated surface and boundary fluxes. Vertical exchanges between surface and deep layers are implicitly captured within ECCO's state estimate but are not explicitly diagnosed in this study. We have now added a clarifying sentence in Section 2 noting this limitation:

"Vertical exchanges between surface and deep Mediterranean layers are implicitly represented in ECCO but not explicitly analyzed in this study." (Line 163)

2 Data & Methods

L119 - River contribution was ignored previously but now attributed to 0.01-0.02 Sv of difference which is about 30-50% of Gibraltar net transport.

To clarify, river runoff was not ignored in our ECCO-based budget analysis. ECCOv4r4 includes river inputs via surface boundary forcing, and these are accounted for in our diagnosed surface freshwater flux (E-P-R). However, when we compared ECCO-derived E-P-R with external satellite-based E and P products (i.e., Figure 2b), we did not include runoff in the satellite estimates due to lack of consistent global runoff data in these products. This naturally results in a small systematic offset (~0.01–0.02 Sv) between ECCO and satellite-derived surface freshwater fluxes.

We also respectfully note that comparing this small offset (~0.01–0.02 Sv) to the long-term mean Gibraltar transport (~0.04 Sv) is not directly meaningful, as this anomaly appears only during certain months and represents a short-term flux discrepancy rather than a persistent or cumulative transport bias.

L140 - It could be at least compared.

As part of addressing the earlier comment, we have now added a comparison with Aydogdu et al. (2023) in Section 2 where basin-mean salinity is discussed (see also response to major comment on salinity comparison). (Line 129)

L157 - Why 320 m? Is this the depth of Mediterranean overflow?

The 320 m depth corresponds to the deepest sill depth at the Strait of Gibraltar that allows inflow and outflow exchange between the Atlantic and Mediterranean in ECCO's model. We have now clarified this in the methods section. (Line 166)

3 Results

Fig. 7 0.05 Sv of "mass gain" is a bit large compared to 0.038 Sv of Soto-Navarro et al. (2010) or other estimates.

We agree that our diagnosed net mass gain of 0.05 Sv is somewhat larger than the 0.038 Sv reported by Soto-Navarro et al. (2010) and other previous estimates. However, it is important to note that our estimate falls within the uncertainty range of ±0.02 Sv, which we explicitly state in the manuscript. This range encompasses the Soto-Navarro et al. value as well as other observational estimates.

We also emphasize that ECCO's estimate reflects a fully observation-constrained, dynamically consistent long-term mean for the 2003–2017 period, which could differ slightly from shorter or earlier observational periods due to interannual variability.

Aydogdu, A., Miraglio, P., Escudier, R., Clementi, E., and Masina, S.: The dynamical role of upper layer salinity in the Mediterranean Sea, in: 7th edition of the Copernicus Ocean State Report (OSR7), edited by: von Schuckmann, K., Moreira, L., Le Traon, P.-Y., Grégoire, M., Marcos, M., Staneva, J., Brasseur, P., Garric, G., Lionello, P., Karstensen, J., and Neukermans, G., Copernicus Publications, State Planet, 1-osr7, 6, https://doi.org/10.5194/sp-1-osr7-6-2023, 2023.

Jarosz, E., W. J. Teague, J. W. Book, and Ş. T. Beşiktepe (2013), Observed volume fluxes and mixing in the Dardanelles Strait, J. Geophys. Res. Oceans, 118, 5007–5021, doi:10.1002/jgrc.20396.

Sanchez-Roman, A., Jorda, G., Sannino, G., and Gomis, D.: Modelling study of transformations of the exchange flows along the Strait of Gibraltar, Ocean Sci., 14, 1547–1566, https://doi.org/10.5194/os-14-1547-2018, 2018

Ünlülata, Ü., Oğuz, T., Latif, M. A., & Özsoy, E. (1990). On the physical oceanography of the Turkish Straits. The physical oceanography of sea straits, 25-60.